# Stacking Variational Bayesian Monte Carlo

**Francesco Silvestrin**                                    *francesco.silvestrin@helsinki.fi*
*Department of Computer Science*
*University of Helsinki*

**Chengkun Li**                                             *chengkun.li@helsinki.fi*
*Department of Computer Science*
*University of Helsinki*

**Luigi Acerbi**                                            *luigi.acerbi@helsinki.fi*
*Department of Computer Science*
*University of Helsinki*

**Reviewed on OpenReview:** *https://openreview.net/forum?id=M2ilYAJdPe&noteId=v2wTqvxlkb*

## Abstract

Approximate Bayesian inference for models with computationally expensive, black-box like-lihoods poses a significant challenge, especially when the posterior distribution is complex. Many inference methods struggle to explore the parameter space efficiently under a limited budget of likelihood evaluations. Variational Bayesian Monte Carlo (VBMC) is a sample-efficient method that addresses this by building a local surrogate model of the log-posterior. However, its conservative exploration strategy, while promoting stability, can cause it to miss important regions of the posterior, such as distinct modes or long tails. In this work, we introduce Stacking Variational Bayesian Monte Carlo (S-VBMC), a method that overcomes this limitation by constructing a robust, global posterior approximation from multiple inde-pendent VBMC runs. Our approach merges these local approximations through a principled and inexpensive post-processing step that leverages VBMC's mixture posterior representa-tion and per-component evidence estimates. Crucially, S-VBMC requires no additional like-lihood evaluations and is naturally parallelisable, fitting seamlessly into existing inference workflows. We demonstrate its effectiveness on two synthetic problems designed to chal-lenge VBMC's exploration and two real-world applications from computational neuroscience, showing substantial improvements in posterior approximation quality across all cases. Our code is available as a Python package at `https://github.com/acerbilab/svbmc`.

## 1 Introduction

Bayesian inference provides a powerful framework for parameter estimation and uncertainty quantification, but it is usually intractable, requiring approximate inference techniques (Brooks et al., 2011; Blei et al., 2017). Many scientific and engineering problems involve black-box models (Sacks et al., 1989; Kennedy & O'Hagan, 2001), where likelihood evaluation is time-consuming and gradients cannot be easily obtained, making traditional approximate inference approaches computationally prohibitive.

A promising approach to tackle expensive likelihoods is to construct a statistical surrogate model that approximates the target distribution, similar in spirit to surrogate approaches to global optimisation using Gaussian processes, generally known as *Bayesian optimisation* (Rasmussen & Williams, 2006; Garnett, 2023). However, unlike Bayesian optimisation, where the goal is to find a single point estimate (the global optimum), here the aim is to reconstruct the shape of the entire posterior distribution. In this setting, attempting to build a single global surrogate model may lead to numerical instabilities and poor approximations when the target distribution is complex or multi-modal, without ad hoc solutions (Wang & Li, 2018; Järvenpää et al.,

2021; Li et al., 2025). Local or constrained surrogate models, while more limited in scope, tend to be more stable and reliable in practice (El Gammal et al., 2023; Järvenpää & Corander, 2024).

Variational Bayesian Monte Carlo (VBMC; Acerbi, 2018) exemplifies this local approach, using active sampling to train a Gaussian process surrogate for the unnormalised log-posterior on which it performs variational inference. VBMC adopts a conservative exploration strategy that yields stable, local approximations (Acerbi, 2019). Compared to other surrogate-based approaches, the method offers a versatile set of features: it returns the approximate posterior as a tractable distribution (a mixture of Gaussians); it provides a lower bound for the model evidence (ELBO) via Bayesian quadrature (Ghahramani & Rasmussen, 2002), useful for model selection; and it can handle noisy log-likelihood evaluations (Acerbi, 2020), which arise in simulation-based models through estimation techniques such as *inverse binomial sampling* (van Opheusden et al., 2020) and *synthetic likelihood* (Wood, 2010; Price et al., 2018). However, VBMC's limited likelihood evaluation budget combined with its local exploration strategy can leave it vulnerable to potentially missing regions of the target posterior – particularly for distributions with distinct modes or long tails.

In this work, we propose a practical, yet effective approach to constructing global surrogate models while overcoming the limitations of standard VBMC by combining multiple local approximations. We introduce *Stacking Variational Bayesian Monte Carlo* (S-VBMC), a method for merging independent VBMC inference runs into a coherent global posterior approximation. S-VBMC inherits its parent algorithm's operating regime and is intended for low- to moderate-dimensional problems ($D \leq 10$). Our approach leverages VBMC's unique properties – its mixture posterior representation and per-component Bayesian quadrature estimates of the ELBO – to combine and reweigh each component through a simple post-processing step. Figure 1 shows an example of two separate posteriors and the combined ("stacked") result obtained with S-VBMC.

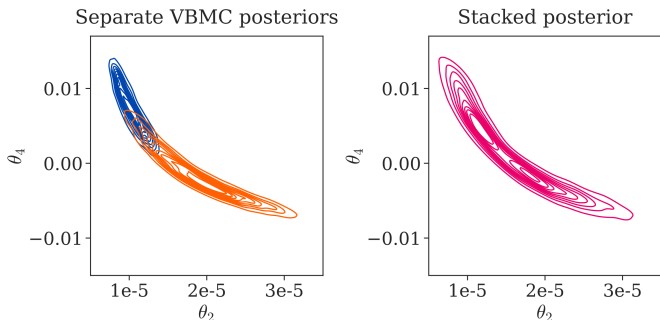

Figure 1: Two separate VBMC posteriors (left, shown as blue and orange contours) and the resulting stacked posterior via S-VBMC (right, red contour) for a neuronal model with real data (see Section 4.5); showing the marginal distribution of two out of the five model parameters.

Crucially, our method requires no additional evaluations of either the original model or the surrogate. This approach is easily parallelisable and naturally fits existing VBMC pipelines that already employ multiple independent runs (Huggins et al., 2023). While our method could theoretically extend to other variational approaches based on mixture posteriors, VBMC is uniquely suitable for it as re-estimation of the ELBO would otherwise become impractical with expensive likelihoods (see Section 3).

**Related work.** Our work addresses the challenge of building global posterior approximations by combining local solutions from the VBMC framework (Acerbi, 2018; 2019; 2020). While the idea of combining posterior distributions has been explored before, previous approaches differ substantially in their goals and methodology.

"Stacking" was first introduced in the context of supervised learning as a method for model averaging (Wolpert, 1992). Given a set of predictive models, the idea was to use a weighted average of their outputs, with the weights optimised to minimise the leave-one-out squared error. This approach has then been

adapted to average Bayesian predictive distributions (Yao et al., 2018; 2022), which the authors named *Bayesian stacking*. This still relies on a leave-one-out strategy to optimise predictive performance, which requires access to the likelihood per data point, while S-VBMC optimises the ELBO on the full dataset, allowing treatment of the log-joint as a black box.

Another relevant approach is *variational boosting* (Guo et al., 2016; Miller et al., 2017; Campbell & Li, 2019). This method builds on black-box variational inference (Ranganath et al., 2014) and entails iteratively running a series of variational optimisations on the whole dataset, with each iteration increasing the complexity (number of components) of a mixture posterior distribution. This allows practitioners to obtain arbitrarily complex (and accurate) posteriors, trading compute time for inference accuracy. However, the process is inherently sequential, whilst S-VBMC can be implemented as a simple post-processing step, allowing individual (VBMC) inference runs to happen in parallel, offering significant computational advantages, further substantiated by VBMC's surrogate-based approach and sample efficiency, which make it particularly suitable for problems with expensive likelihoods.

Parallel computations have, on the other hand, been leveraged in a number of "divide-and-conquer" or *embarrassingly parallel* approximate inference techniques, starting from embarrassingly parallel Markov Chain Monte Carlo (MCMC) (Neiswanger et al., 2014). All these methods are based on dividing the data into subsets which are processed separately to obtain a set of "sub-posteriors", which are then merged in various ways to recreate the full (approximate) posterior (Wang & Dunson, 2013; Wang et al., 2015; Nemeth & Sherlock, 2018; Srivastava et al., 2018; Scott et al., 2022; De Souza et al., 2022; Chan et al., 2023). These methods are mostly motivated by the need to process very large datasets (Scott et al., 2022), or by privacy concerns requiring federated learning (Liang et al., 2025). In contrast, S-VBMC is motivated by a need to capture complex posteriors that elude single inference runs. Therefore, individual runs all use the complete dataset. This allows our method not to rely on the quality and representativeness of the sub-datasets, and to remain robust to individual run failures.

**Outline.** We first introduce variational inference and VBMC (Section 2), then present our algorithm for stacking VBMC posteriors (Section 3). We demonstrate the effectiveness of our approach through experiments on two synthetic problems and two real-world applications that are challenging for VBMC (Section 4). We then discuss an observed bias buildup in the ELBO estimation and propose a practical heuristic to counteract it (Section 5). We finally discuss our results (Section 6) and conclude with closing remarks (Section 7). Appendix A contains supplementary materials.

## 2 Background

### 2.1 Bayesian Inference

Given a dataset $\mathcal{D}$, and a model parametrised by the vector $\boldsymbol{\theta} \in \mathbb{R}^D$ describing how $\mathcal{D}$ was generated, Bayesian inference represents a principled framework to infer the probability distributions over $\boldsymbol{\theta}$. This is achieved through Bayes' rule:

$$p(\boldsymbol{\theta} \mid \mathcal{D}) = \frac{p(\boldsymbol{\theta})p(\mathcal{D} \mid \boldsymbol{\theta})}{p(\mathcal{D})}, \tag{1}$$

where $p(\boldsymbol{\theta})$ represents the prior over model parameters, $p(\mathcal{D} \mid \boldsymbol{\theta})$ the likelihood and $p(\mathcal{D}) = \int p(\boldsymbol{\theta})p(\mathcal{D} \mid \boldsymbol{\theta})d\boldsymbol{\theta}$ the normalising constant, also called marginal likelihood or model evidence. This latter quantity is particularly useful for Bayesian model selection (MacKay, 2003), but the integral is often intractable, requiring some approximation technique (Brooks et al., 2011; Blei et al., 2017). In the following section, we discuss one of such techniques.

### 2.2 Variational Inference

Considering the setup outlined in Eq. 1, variational inference (Blei et al., 2017) is a technique that approximates the true posterior $p(\boldsymbol{\theta}|\mathcal{D})$ with a parametric distribution $q_{\boldsymbol{\phi}}(\boldsymbol{\theta})$, where $\boldsymbol{\phi}$ denotes the optimisable

variational parameters. This is achieved by maximising the evidence lower bound (ELBO):

$$\text{ELBO}(\boldsymbol{\phi}) = \mathbb{E}_{q_{\boldsymbol{\phi}}} \left[ \log p(\mathcal{D}|\boldsymbol{\theta})p(\boldsymbol{\theta}) \right] + \mathcal{H} \left[ q_{\boldsymbol{\phi}}(\boldsymbol{\theta}) \right], \tag{2}$$

where the first term is the expected log-joint distribution (the joint being likelihood times prior) and the second term is the entropy of the variational posterior. Maximising Eq. 2 is equivalent to minimising the Kullback-Leibler divergence between $q_{\boldsymbol{\phi}}(\boldsymbol{\theta})$ and the true posterior, as

$$\begin{aligned} D_{\text{KL}}[q_{\boldsymbol{\phi}}(\boldsymbol{\theta}) \,||\, p(\boldsymbol{\theta} \mid \mathcal{D})] &= \mathbb{E}_{q_{\boldsymbol{\phi}}} \left[ \log \frac{q_{\boldsymbol{\phi}}(\boldsymbol{\theta})}{p(\boldsymbol{\theta} \mid \mathcal{D})} \right] \\ &= -\mathbb{E}_{q_{\boldsymbol{\phi}}}[\log p(\boldsymbol{\theta}, \mathcal{D})] + \mathbb{E}_{q_{\boldsymbol{\phi}}}[\log p(\mathcal{D})] + \mathbb{E}_{q_{\boldsymbol{\phi}}}[\log q_{\boldsymbol{\phi}}(\boldsymbol{\theta})], \end{aligned} \tag{3}$$

and, since the model evidence $p(\mathcal{D})$ is already a constant,

$$\begin{aligned} \log p(\mathcal{D}) - D_{\text{KL}}[q_{\boldsymbol{\phi}}(\boldsymbol{\theta}) \,||\, p(\boldsymbol{\theta} \mid \mathcal{D})] &= \mathbb{E}_{q_{\boldsymbol{\phi}}}[\log p(\boldsymbol{\theta}, \mathcal{D})] + \mathcal{H} \left[ q_{\boldsymbol{\phi}}(\boldsymbol{\theta}) \right] \\ &= \text{ELBO}(\boldsymbol{\phi}). \end{aligned} \tag{4}$$

Crucially, since $D_{\text{KL}}[q \,||\, p] \geq 0$, the ELBO provides a lower bound on the log model evidence $\log p(\mathcal{D})$ (hence the name), with equality when the approximation matches the true posterior (*i.e.*, when $D_{\text{KL}}[q \,||\, p] = 0$), thus constituting a useful metric for model selection.

### 2.3 Variational Bayesian Monte Carlo (VBMC)

VBMC (Acerbi, 2018; 2020) is a sample-efficient technique to obtain a variational approximation of a target density with only a small number of likelihood evaluations, often of the order of a few hundred. VBMC uses a Gaussian process (GP) as a surrogate of the log-joint, Bayesian quadrature to calculate the expected log-joint, and active sampling to decide which parameters to evaluate next. As an in-depth knowledge of the inner workings of VBMC is not necessary to understand our core contribution, here we only describe its relevant aspects. An interested reader should refer to the original papers (Acerbi, 2018; 2020) or Appendix A.1, where we provide a more detailed description of the algorithm.

As mentioned above, VBMC uses a GP as a surrogate of the target, which is often an unnormalised log-posterior (*i.e.*, the log-joint). Crucially, VBMC performs variational inference on the surrogate

$$f(\boldsymbol{\theta}) \approx \log p(\mathcal{D}|\boldsymbol{\theta})p(\boldsymbol{\theta}) \tag{5}$$

instead of the true, expensive model. The efficacy of VBMC hinges on the quality of such approximation.

The variational posterior in VBMC is defined as a flexible mixture of $K$ components,

$$q_{\boldsymbol{\phi}}(\boldsymbol{\theta}) = \sum_{k=1}^{K} w_k q_{k,\boldsymbol{\phi}}(\boldsymbol{\theta}), \tag{6}$$

where $q_k$ is the $k$-th component (a multivariate normal) and $w_k$ its mixture weight, with $\sum_{k=1}^{K} w_k = 1$ and $w_k \geq 0$. Plugging in the mixture posterior, the ELBO (Eq. 2) becomes:

$$\text{ELBO}(\boldsymbol{\phi}) = \sum_{k=1}^{K} w_k \mathbb{E}_{q_{k,\boldsymbol{\phi}}} \left[ \log p(\mathcal{D}|\boldsymbol{\theta})p(\boldsymbol{\theta}) \right] + \mathcal{H} \left[ q_{\boldsymbol{\phi}}(\boldsymbol{\theta}) \right] = \sum_{k=1}^{K} w_k I_k + \mathcal{H} \left[ q_{\boldsymbol{\phi}}(\boldsymbol{\theta}) \right] \tag{7}$$

where we defined the $k$-th component of the expected log-joint as:

$$I_k = \mathbb{E}_{q_{k,\boldsymbol{\phi}}} \left[ \log p(\mathcal{D}|\boldsymbol{\theta})p(\boldsymbol{\theta}) \right] \approx \mathbb{E}_{q_{k,\boldsymbol{\phi}}} \left[ f(\boldsymbol{\theta}) \right] = \hat{I}_k. \tag{8}$$

The efficacy of VBMC stems from the fact that Eq. 8 has a closed-form Gaussian expression via Bayesian quadrature (O'Hagan, 1991; Ghahramani & Rasmussen, 2002), which yields posterior mean $\hat{I}_k$ and covariance matrix $J_{kk'}$ (Acerbi, 2018). The entropy of a mixture of Gaussians does not have an analytical solution, but

gradients can be estimated via Monte Carlo. Thus, using the posterior mean of Eq. 8 as a plug-in estimator for the expected log-joint of each component, Eq. 7 can be efficiently optimised via stochastic gradient ascent (Kingma & Ba, 2014).

VBMC differs from other approaches that directly try to apply Bayesian quadrature to solve Bayes' rule (*e.g.*, Ghahramani & Rasmussen, 2002; Osborne et al., 2012; Gunter et al., 2014; Adachi et al., 2022), in that instead it leverages Bayesian quadrature to estimate the ELBO. *This is a simpler problem*, since instead of attempting to solve the *global* integral of Eq. 1, namely the integral of prior times likelihood, where the prior is often diffuse, it mainly deals with the *local* integrals in Eq. 7, namely the expected log-joint, *i.e.*, the integral of the approximate posterior times log-joint, where the approximate posterior is under our control and often more localised, and the entropy, which can often be estimated or approximated for known distributions.

## 3 Stacking VBMC

In this work, we introduce *Stacking VBMC* (S-VBMC), a novel approach to merge different variational posteriors obtained from different runs on the same model and dataset. Crucially, these runs can happen in parallel, with no information exchange required. The core idea is that a single *global* surrogate of the log-joint $f(\boldsymbol{\theta})$ might be inaccurate in some regions of the posterior. Therefore, it would be beneficial to leverage the combination of $M$ *local* surrogates $\{f_m(\boldsymbol{\theta})\}_{m=1}^{M}$ instead, each yielding a good approximation of the target in a different parameter region.

### 3.1 Optimisation objective

Given $M$ independent VBMC runs, one obtains $M$ variational posteriors

$$q_{\boldsymbol{\phi}_m}(\boldsymbol{\theta}) = \sum_{k=1}^{K_m} w_{m,k} q_{k,\boldsymbol{\phi}_m}(\boldsymbol{\theta}), \tag{9}$$

each with $K_m$ Gaussian components, as well as $M$ different $\hat{\mathbf{I}}_m$ vectors, as per Eq. 8, with $\hat{\mathbf{I}}_m = (\hat{I}_{m,1}, \ldots, \hat{I}_{m,K_m})$. Our approach consists of "stacking" the Gaussian components of all posteriors $q_{\boldsymbol{\phi}_m}(\boldsymbol{\theta})$, leaving all individual components' parameters (means and covariances) unchanged, and reoptimising *all* the weights. The full set of new mixture weights is denoted by the vector $\tilde{\mathbf{w}} = \{\tilde{w}_{m,k}\}_{m=1,k=1}^{M,K_m}$. The complete set of parameters for the final stacked posterior, $q_{\tilde{\boldsymbol{\phi}}}$, is denoted by $\tilde{\boldsymbol{\phi}}$. This set consists of the frozen parameters (all means and covariances) from the original components $\{q_{k,\boldsymbol{\phi}_m}\}$ and the newly optimised weights $\tilde{\mathbf{w}}$.

Thus, given the stacked posterior

$$q_{\tilde{\boldsymbol{\phi}}}(\boldsymbol{\theta}) = \sum_{m=1}^{M} \sum_{k=1}^{K_m} \tilde{w}_{m,k} q_{k,\boldsymbol{\phi}_m}(\boldsymbol{\theta}), \tag{10}$$

we optimise the global evidence lower bound with respect to the weights $\tilde{\mathbf{w}}$,

$$\text{ELBO}_{\text{stacked}}(\tilde{\mathbf{w}}) = \sum_{m=1}^{M} \sum_{k=1}^{K_m} \tilde{w}_{m,k} \hat{I}_{m,k} + \mathcal{H}\left[q_{\tilde{\boldsymbol{\phi}}}(\boldsymbol{\theta})\right]. \tag{11}$$

It should be noted that each VBMC run applies its own parameter transformation $g_m(\cdot)$ during inference (Acerbi, 2020), so variational posteriors are returned in transformed coordinates $q_{\boldsymbol{\psi}_m}(g_m(\boldsymbol{\theta}))$. For clarity, in this section, we present all quantities in a common parameter space of $\boldsymbol{\theta}$ and VBMC and S-VBMC posteriors expressed accordingly as $q_{\boldsymbol{\phi}_m}(\boldsymbol{\theta})$, and $q_{\tilde{\boldsymbol{\phi}}}(\boldsymbol{\theta})$, respectively. In practice, when applying our stacking approach, we account for these per–run transformations via change-of-variables (Jacobian) corrections; see Appendix A.2 for details.

### 3.2 Algorithm

As the entropy term in Eq. 11 does not have a closed-form solution, it needs to be estimated via Monte Carlo sampling. To do this, we take $S$ samples $\left\{ \mathbf{x}_{m,k}^{(s)} \sim q_{k,\phi_m} \right\}_{s=1}^{S}$ from each component of the stacked posterior, and estimate the entropy as

$$\mathcal{H}\left[ q_{\tilde{\phi}}(\boldsymbol{\theta}) \right] \approx - \sum_{m=1}^{M} \sum_{k=1}^{K_m} \tilde{w}_{m,k} \left( \frac{1}{S} \sum_{s=1}^{S} \log q_{\tilde{\phi}}\left( \mathbf{x}_{m,k}^{(s)} \right) \right). \tag{12}$$

This works because

$$\begin{aligned}
\mathcal{H}\left[ q_{\tilde{\phi}}(\boldsymbol{\theta}) \right] &= -\mathbb{E}_{q_{\tilde{\phi}}} \left[ \log q_{\tilde{\phi}}(\boldsymbol{\theta}) \right] \\
&= -\int q_{\tilde{\phi}}(\boldsymbol{\theta}) \log q_{\tilde{\phi}}(\boldsymbol{\theta}) d\boldsymbol{\theta} \\
&= -\int \sum_{m=1}^{M} \sum_{k=1}^{K_m} \tilde{w}_{m,k} q_{k,\phi_m}(\boldsymbol{\theta}) \log q_{\tilde{\phi}}(\boldsymbol{\theta}) d\boldsymbol{\theta} \\
&= -\sum_{m=1}^{M} \sum_{k=1}^{K_m} \tilde{w}_{m,k} \int q_{k,\phi_m}(\boldsymbol{\theta}) \log q_{\tilde{\phi}}(\boldsymbol{\theta}) d\boldsymbol{\theta} \\
&= -\sum_{m=1}^{M} \sum_{k=1}^{K_m} \tilde{w}_{m,k} \mathbb{E}_{q_{k,\phi_m}} \left[ \log q_{\tilde{\phi}}(\boldsymbol{\theta}) \right],
\end{aligned} \tag{13}$$

where we approximate the expectation via Monte Carlo,

$$\mathbb{E}_{q_{k,\phi_m}} \left[ \log q_{\tilde{\phi}}(\boldsymbol{\theta}) \right] \approx \frac{1}{S} \sum_{s=1}^{S} \log q_{\tilde{\phi}}\left( \mathbf{x}_{m,k}^{(s)} \right). \tag{14}$$

Importantly, with this formulation, the entropy is differentiable with respect to the weights $\tilde{\mathbf{w}}$.

Since weights must be non-negative and sum to one (*i.e.*, they live on the probability simplex), for the purpose of optimisation we follow standard practices in computational statistics (e.g., Carpenter et al., 2017): we reparameterise the weights with unconstrained logits $\mathbf{a}$ and compute the objective with softmax weights,

$$\tilde{w}_{m,k} = \frac{\exp(a_{m,k})}{\sum_{m'=1}^{M} \sum_{k'=1}^{K_{m'}} \exp(a_{m',k'})}. \tag{15}$$

We initialise the logits as

$$a_{m,k} = \log(w_{m,k}) + \mathrm{ELBO}(\phi_m) - \max_{\substack{1 \le m' \le M \\ 1 \le k' \le K_{m'}}} \left( \log(w_{m',k'}) + \mathrm{ELBO}(\phi_{m'}) \right), \tag{16}$$

where the last term ensures that $\max(\mathbf{a}) = 0$, preventing underflow in the exponentials. This is equivalent to initialising the weights as

$$\tilde{w}_{m,k} \propto w_{m,k} \exp(\mathrm{ELBO}(\phi_m)), \tag{17}$$

effectively weighting each individual posterior by its approximate evidence (*i.e.*, the exponential of the corresponding ELBO). This assigns a higher initial weight to the components coming from the best runs, providing a good starting point for the optimisation process.

We can then use the Adam optimiser (Kingma & Ba, 2014) to optimise the stacked ELBO. An overview of the procedure is outlined in Algorithm 1.

Notably, the optimisation can be performed as a pure post-processing step, requiring neither evaluations of the original likelihood $p(\mathcal{D}|\boldsymbol{\theta})$ nor of the surrogate models $f_m$, only that the estimates $\hat{I}_{m,k}$ are stored, as in current implementations (Huggins et al., 2023).

---

**Algorithm 1** S-VBMC

---

**Input**: outputs of $M$ VBMC runs $\{\boldsymbol{\phi}_m, \hat{\mathbf{I}}_m, \text{ELBO}(\boldsymbol{\phi}_m)\}_{m=1}^M$

1. Stack the posteriors to obtain a mixture of $\sum_{m=1}^M K_m$ Gaussians

2. Reparametrise the weights with unconstrained logits $\mathbf{a}$

3. Initialise the logits as in Eq. 16

4. **repeat**

    (a) Compute the normalised weights $\tilde{\mathbf{w}}$ as in Eq. 15
    (b) Estimate $\text{ELBO}_{\text{stacked}}(\tilde{\mathbf{w}})$ as in Eq. 11 with the entropy approximated as in Eq. 12
    (c) Compute $\frac{d\text{ELBO}_{\text{stacked}}(\mathbf{a})}{d\mathbf{a}}$
    (d) Update $\mathbf{a}$ with a step of Adam (Kingma & Ba, 2014)

    **until** $\text{ELBO}_{\text{stacked}}(\tilde{\mathbf{w}})$ converges

**return** $\tilde{\boldsymbol{\phi}}$, $\text{ELBO}_{\text{stacked}}$

---

Our stacking method hinges on the key feature of VBMC of providing accurate estimates $\hat{I}_{m,k}$. While in principle Eq. 11 could apply to any collection of variational posterior mixtures, without an efficient way of calculating each $I_k$ (Eq. 8), optimisation of the stacked ELBO would require many likelihood evaluations, which would be prohibitive for problems with expensive, black-box likelihoods.

In the following, we demonstrate the efficacy of this approach.

## 4 Experiments

In this section, we describe our experimental procedure (Section 4.1), baselines (Section 4.2), evaluation metrics (Section 4.3), and results for synthetic (Section 4.4) and real-world problems (Section 4.5). Finally, we briefly discuss computational costs in Section 4.6. Additional results are reported in the appendix, including additional experiments (Appendix A.3), more extensive tables of results (Appendix A.4), and example visualisations of posterior approximations (Appendix A.5).

### 4.1 Procedure

We first tested our method on two synthetic problems, designed to be particularly challenging for VBMC, and then on two real-world datasets and models (see below for full descriptions). We considered both noiseless problems (exact estimation) and noisy problems where Gaussian noise with $\sigma = 3$ is applied to each log-likelihood measurement, emulating what practitioners might find when estimating the likelihood via simulation (van Opheusden et al., 2020). For each benchmark, we considered 100 VBMC runs obtained with the `pyvbmc` Python package (Huggins et al., 2023) that satisfied the following conditions:

1. the algorithm had converged (as assessed by the `pyvbmc` software);

2. $\max_{1 \leq k \leq K}(J_{k,k}) < 5$, so all the estimates $\hat{I}_k$ had an associated variance lower than 5.

The reason for the latter is that S-VBMC's efficacy is strongly dependent on accurate VBMC estimates of the real $I_k$ from Bayesian quadrature (see Sections 2.3 and 3.2), and we sought to filter out "poorly converged" runs where this might not be the case. The results of this filtering procedure can be found in Appendix A.4.1. We performed all VBMC runs using the default settings (which always resulted in posteriors with 50 components, $K_m = 50 \; \forall \; m \in \{1, ..., M\}$) and random uniform initialisation within plausible parameter bounds (Acerbi, 2018). To investigate the effect of combining a different number of posteriors, we adopted a bootstrapping approach: from these 100 runs, we randomly sampled and stacked

with S-VBMC a varying number of runs (between 2 and 40) twenty times each, and computed the median with corresponding 95% confidence interval (computed from 10000 bootstrap resamples) for all metrics, described below. For all benchmark problems, the entropy is approximated as in Eq. 12 with $S = 20$ during optimisation, and a final estimation (reported in all tables and figures) is performed with $S = 100$ after convergence. The ELBO is optimised using Adam (Kingma & Ba, 2014) with learning rate set to 0.1.

All the experiments presented in this work were run on an AMD EPYC 7452 Processor with 16GB of RAM.

## 4.2 Baseline methods

**Black-box variational inference.** We used black-box variational inference (BBVI; Ranganath et al., 2014) as a baseline for all our benchmark problems.

Our implementation follows Li et al. (2025). For gradient-free black-box models, we cannot use the reparameterisation trick (Kingma & Welling, 2013) to estimate ELBO gradients. Instead, we employ the score function estimator (REINFORCE; Ranganath et al., 2014) with control variates to reduce gradient variance.

The variational posterior is parameterised as a mixture of Gaussians (MoG) with either $K = 50$ or $K = 500$ components, matching the form used in VBMC. We initialise the component means near the origin by adding Gaussian noise ($\sigma = 0.1$) and set all component variances to 0.01. We optimise the ELBO using Adam (Kingma & Ba, 2014) with stochastic gradients, performing a grid search over Monte Carlo sample sizes $\{1, 10, 100\}$ and learning rates $\{0.01, 0.001\}$. We select the best hyperparameters based on the estimated ELBO.

For a fair comparison with S-VBMC, we set the target evaluation budget for a BBVI run to $2000(D+2)$ and $3000(D+2)$ evaluations for noiseless and noisy problems, respectively, matching the maximum evaluations used by 40 VBMC runs in total.

**Naive Stacking.** As a further baseline, we implemented a "naive" version of our stacking approach, consisting of a simple averaging of the individual VBMC posteriors. For this, we rewrite the stacked posterior as

$$q_{\tilde{\phi}}(\boldsymbol{\theta}) = \sum_{m=1}^{M} \tilde{\omega}_m q_{\phi_m}(\boldsymbol{\theta}) \tag{18}$$

and simply set

$$\tilde{\omega}_1 = \tilde{\omega}_2 = ... = \tilde{\omega}_m = 1/M. \tag{19}$$

We call this approach "Naive Stacking" (NS).

As for S-VBMC, we ran NS by randomly sampling a varying number of runs (between 2 and 40) twenty times each, and then computed the median with corresponding 95% confidence intervals for all metrics.

## 4.3 Metrics

Following Acerbi (2020); Li et al. (2025), we evaluate our method using three metrics:

1. The absolute difference between true and estimated log marginal likelihood ($\Delta$LML), where values $< 1$ are considered negligible for model selection (Burnham & Anderson, 2003).

2. The mean marginal total variation distance (MMTV), which measures the average (lack of) overlap between true and approximate posterior marginals across dimensions:

$$\mathrm{MMTV}(p, q) = \frac{1}{2D} \sum_{d=1}^{D} \int_{-\infty}^{\infty} |p_d(x_d) - q_d(x_d)| \, dx_d, \tag{20}$$

where $p_d$ and $q_d$ denote the marginal distributions along the $d$-th dimension.

3. The "Gaussianised" symmetrised KL divergence (GsKL), which evaluates differences in means and covariances between the approximate and true posterior:

$$\text{GsKL}(p, q) = \frac{1}{2D} \left[ D_{\text{KL}}\left(\mathcal{N}[p]||\mathcal{N}[q]\right) + D_{\text{KL}}(\mathcal{N}[q]||\mathcal{N}[p]) \right],$$ (21)

where $\mathcal{N}[p]$ denotes a Gaussian with the same mean and covariance as $p$.

We consider MMTV $< 0.2$ and GsKL $< \frac{1}{8}$ as target thresholds for reasonable posterior approximation (Li et al., 2025). Ground-truth estimates of the log marginal likelihood and posterior distributions are obtained through numerical integration, extensive MCMC sampling, or analytical methods as appropriate for each problem.

### 4.4 Synthetic problems

**GMM target.** Our synthetic GMM target consists of a mixture of 20 bivariate Gaussian components arranged in four distinct clusters. The cluster centroids were positioned at $(-8, -8)$, $(-7, 7)$, $(6, -6)$ and $(5, 5)$. Around each centroid, we placed five Gaussian components with means drawn from $\mathcal{N}(\boldsymbol{\mu}_c, \mathbf{I})$, where $\boldsymbol{\mu}_c$ is the respective cluster centroid and $\mathbf{I}$ is the $2\times2$ identity matrix. Each component was assigned unit marginal variances and a correlation coefficient of $\pm0.5$ (randomly selected with equal probability). This configuration produces an irregular mixture structure that requires a substantial number of components to approximate accurately. All components were assigned equal mixing weights. The resulting distribution is illustrated in Figure 2 (top panels).

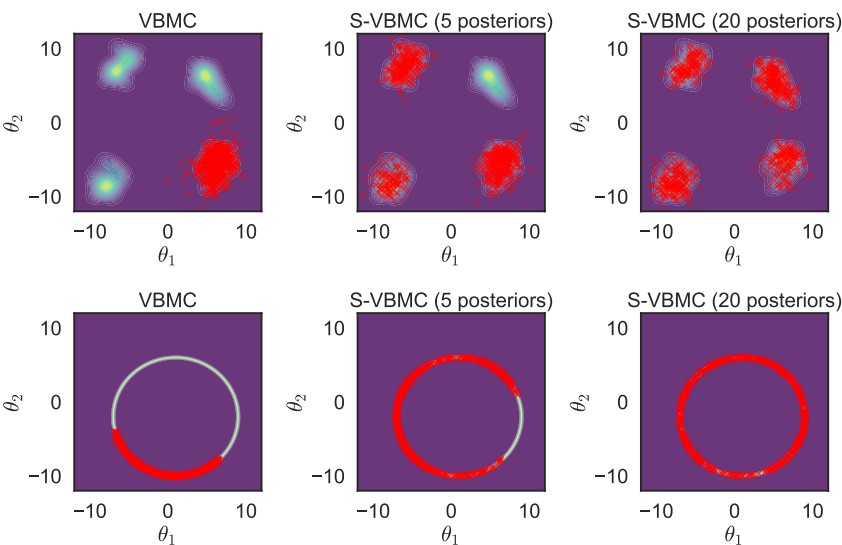

Figure 2: Examples of overlap between the ground truth and the posterior when combining different numbers of VBMC runs on the GMM (top panels) and ring (bottom panels) synthetic benchmarks. The red points indicate samples from the posterior approximation, with the target density depicted with colour gradients in the background.

In our experiments, we used both a noiseless version of this target (*i.e.*, exact target evaluation) and a noisy one, where we applied i.i.d. Gaussian noise ($\sigma = 3$) to each log-likelihood evaluation.

**Ring target.** Our second synthetic target is a ring-shaped distribution defined by the probability density function

$$p_{\text{ring}}(\theta_1, \theta_2) \propto \exp\left(-\frac{(r - R)^2}{2\sigma^2}\right)$$ (22)

where $r = \sqrt{(\theta_1 - c_1)^2 + (\theta_2 - c_2)^2}$ represents the radial distance from centre $(c_1, c_2)$, $R$ is the ring radius, and $\sigma$ controls the width of the annulus. We set $R = 8$, $\sigma = 0.1$, and centred the ring at $(c_1, c_2) = (1, -2)$. The small value of $\sigma$ produces a narrow annular distribution that challenges VBMC's exploration capabilities. The resulting distribution is shown in Figure 2 (bottom panels).

As with the GMM target, we used both a noiseless version of this benchmark and a noisy one ($\sigma = 3$) in our experiments.

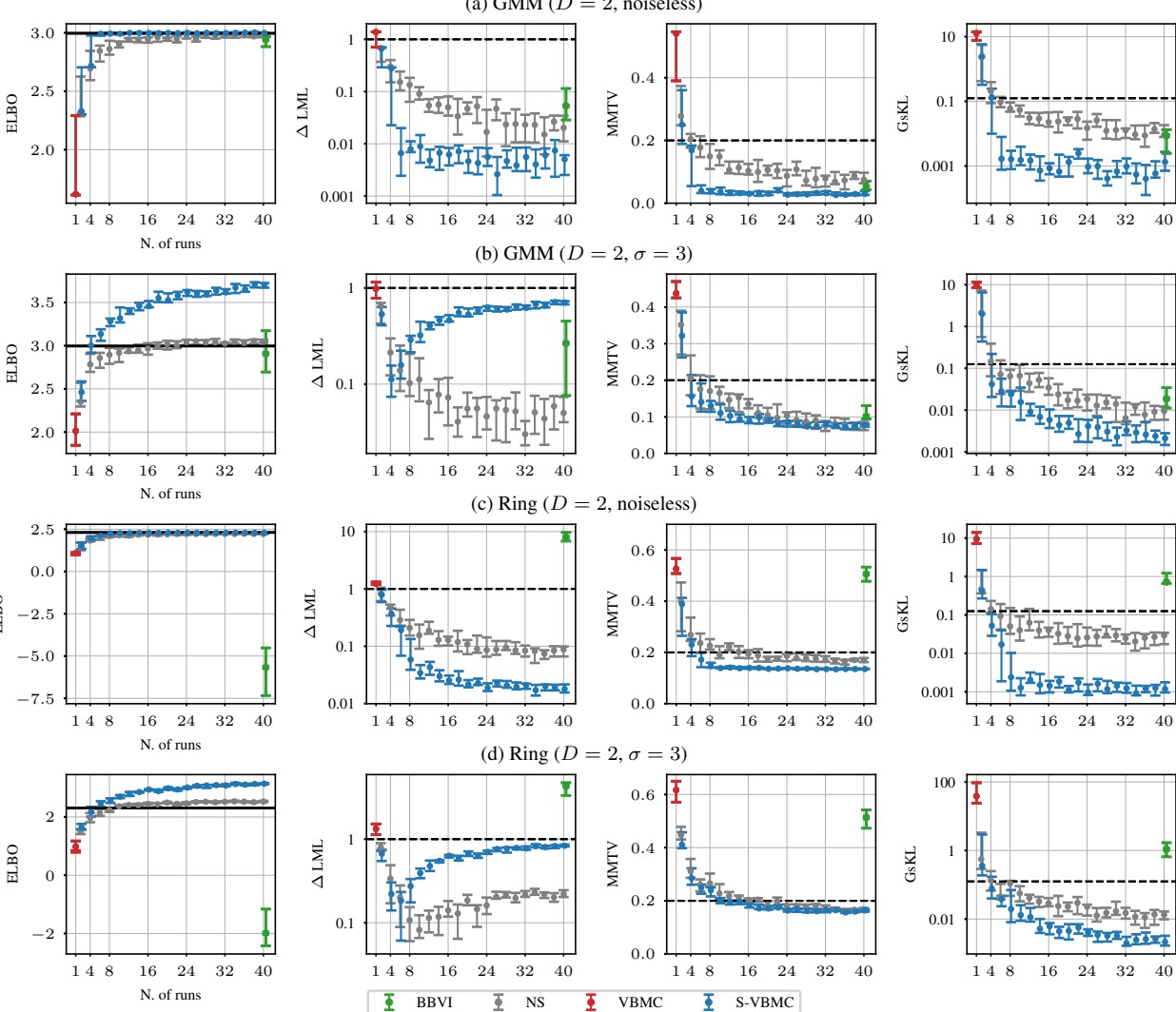

Figure 3: Synthetic problems. Metrics plotted as a function of the number of VBMC runs stacked (median and 95% confidence interval). Metrics are plotted for S-VBMC (blue), VBMC (red), and NS (grey). The best BBVI results are shown in green. The black horizontal line in the ELBO panels represents the ground-truth LML, while the dashed lines on $\Delta$LML, MMTV, and GsKL denote desirable thresholds for each metric (good performance is below the threshold; see Section 4.3)

**Results.** Results in Figure 3 and Table A.3 show that merging more posteriors leads to a steady improvement in the GsKL and MMTV metrics, which measure the quality of the posterior approximation. Remarkably, S-VBMC proves to be robust to noisy targets, with minor differences between noiseless and noisy settings. In all our synthetic benchmarks (both in noiseless and noisy settings), we observe that the values of the GsKL and MMTV metrics, which directly compare to the ground-truth target densities, reach good values – and start to plateau, or at least to improve at a much slower pace – by the time 10 VBMC

posteriors are stacked. This suggests that a value of $M = 10$ (or even $M \approx 6$ for noiseless problems, where metrics tend to converge faster) would be sufficient to obtain accurate stacked posteriors.

S-VBMC outperforms the BBVI baseline on the ring-shaped synthetic target. The BBVI baseline performs well and is only marginally worse compared to S-VBMC only on the GMM problem, where it effectively managed to capture the four clusters (see Figure 2 for a visualisation). While NS exhibits a similar improvement pattern with increasing values of $M$, S-VBMC consistently outperforms it, with differences being larger in noiseless settings.

As expected by design, individual VBMC runs tended to explore the two synthetic target distributions only partially, leading to poor performance. Still, the random initialisations allowed different runs to discover different portions of the posterior, allowing the merging process to cover the whole target (see Figure 2).

Finally, we observe that, in noisy settings, while the ELBO keeps increasing, the $\Delta$LML error (difference between ELBO and true log marginal likelihood) initially decreases but then increases again as further components are added, a point which we will discuss later.

### 4.5 Real-world problems.

**Neuronal model.** Our first real-world problem involved fitting five biophysical parameters of a detailed compartmental model of a hippocampal CA1 pyramidal neuron. The model was constructed based on experimental data comprising a three-dimensional morphological reconstruction and electrophysiological recordings of neuronal responses to current injections. The deterministic neuronal responses were simulated using the NEURON simulation environment (Hines & Carnevale, 1997; Hines et al., 2009), applying current step inputs that matched the experimental protocol. The model's parameters characterise key biophysical properties: intracellular axial resistivity ($\theta_1$), leak current reversal potential ($\theta_2$), somatic leak conductance ($\theta_3$), dendritic conductance gradient ($\theta_4$, per $\mu$m), and a dendritic surface scaling factor ($\theta_5$). Based on independent measurements of membrane potential fluctuations, observation noise was modelled as a stationary Gaussian process with zero mean and a covariance function estimated from the data. The covariance structure was captured by the product of a cosine and an exponentially decaying function. For a similar approach applied to cerebellar Golgi cells, see Szoboszlay et al. (2016).

This model allowed exact log-likelihood evaluations, and no noise was added.

**Multisensory causal inference model.** Perceptual causal inference involves determining whether multiple sensory stimuli originate from a common source, a problem of particular interest in computational cognitive neuroscience (Körding et al., 2007). Our second real-world problem involved fitting a visuo-vestibular causal inference model to empirical data from a representative participant (S1 from Acerbi et al., 2018). In each trial of the modelled experiment, participants seated in a moving chair reported whether they perceived their movement direction ($s_{\text{vest}}$) as congruent with an experimentally-manipulated looming visual field ($s_{\text{vis}}$). The model assumes participants receive noisy sensory measurements, with vestibular information $z_{\text{vest}} \sim \mathcal{N}(s_{\text{vest}}, \sigma_{\text{vest}}^2)$ and visual information $z_{\text{vis}} \sim \mathcal{N}(s_{\text{vis}}, \sigma_{\text{vis}}^2(c))$, where $\sigma_{\text{vest}}^2$ and $\sigma_{\text{vis}}^2$ represent sensory noise variances. The visual coherence level $c$ was experimentally manipulated across three levels ($c_{\text{low}}$, $c_{\text{med}}$, $c_{\text{high}}$). The model assumes participants judge the stimuli as having a common cause when the absolute difference between sensory measurements falls below a threshold $\kappa$, with a lapse rate $\lambda$ accounting for random responses. The model parameters $\boldsymbol{\theta}$ comprise the visual noise parameters $\sigma_{\text{vis}}(c_{\text{low}})$, $\sigma_{\text{vis}}(c_{\text{med}})$, $\sigma_{\text{vis}}(c_{\text{high}})$, vestibular noise $\sigma_{\text{vest}}$, lapse rate $\lambda$, and decision threshold $\kappa$ (Acerbi et al., 2018).

We fitted this model assuming log-likelihood measurement noise ($\sigma = 3$, which we applied to each log-likelihood evaluation).

**Results.** The results in Figure 4 and Table A.4 confirm our earlier findings of improvements across the posterior metrics. We also find that S-VBMC is robust to noisy targets for real data, with performance that improves with the increasing number of stacked runs in the multisensory model problem.

Similar to what we observed for the synthetic problems, we find that stacking $\approx$10 VBMC posteriors seems to be sufficient to vastly improve posterior quality (as indexed by the GsKL and MMTV metrics), compared to

a single VBMC run ($M = 1$), with relatively small additional improvements for $M > 10$. The only exception to this is the GsKL metric in the neuronal model, where the lower confidence intervals start to go below the desirable threshold (see Section 4.3) only if at least 18 posteriors are merged, and the full confidence interval never fully stabilises below such a threshold. However, this was our most challenging benchmark problem, as evidenced by the very poor performance obtained by both single-run VBMC and BBVI. Thus, while S-VBMC here does not achieve a posterior approximation fully on par with gold-standard MCMC (used to obtain the ground-truth posterior in this problem), the stacking process still vastly improves posterior quality at a much lower computational cost.

Finally, similarly to the synthetic problems, we see that S-VBMC performs consistently better than standard VBMC and BBVI across all metrics in both scenarios. Despite similar improvement patterns with increasing values of $M$, as observed above, S-VBMC consistently outperforms NS, with starker differences in the noisy setting (*i.e.*, the multisensory model).

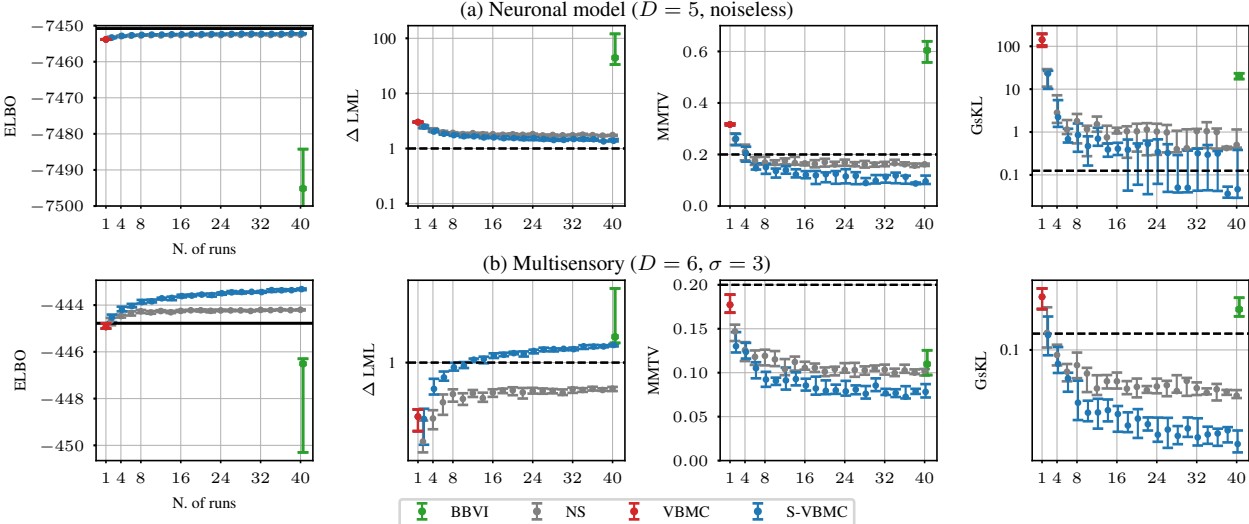

Figure 4: Real-world problems. Metrics plotted as a function of the number of VBMC runs stacked (median and 95% confidence interval). Metrics are plotted for S-VBMC (blue), VBMC (red), and NS (grey). The best BBVI results are shown in green. The black horizontal line in the ELBO panels represents the ground-truth LML, while the dashed lines on ΔLML, MMTV, and GsKL denote desirable thresholds for each metric (good performance is below the threshold; see Section 4.3). The BBVI error bar in the plot displaying the ELBO in the neuronal model (top left) is truncated for clarity.

## 4.6   Computational overhead

Here we present details about the additional computational cost (quantified as compute time) introduced by S-VBMC on top of VBMC. Additional runtime analyses can be found in Appendix A.3.2.

Figure 5 illustrates how S-VBMC introduces a relatively small computational overhead, even when comparing the post-process cost of S-VBMC with the average cost of *one* VBMC run, under the idealised condition where the $M$ VBMC runs happen all in parallel.[1] In particular, running our algorithm with $M \approx 10$ – which vastly improves the resulting posterior, as shown above and in Appendices A.4.2 and A.5 – adds a small amount of post-processing time to VBMC for all our benchmark problems ($\approx$ 5-15% overhead).

Put together, our results confirm that S-VBMC yields high returns in terms of inference performance at a very marginal cost in terms of compute time.

---

[1]In practice, completing $M$ VBMC runs will be more expensive due to additional parallelisation costs, making S-VBMC's relative overhead even smaller than what we report here.

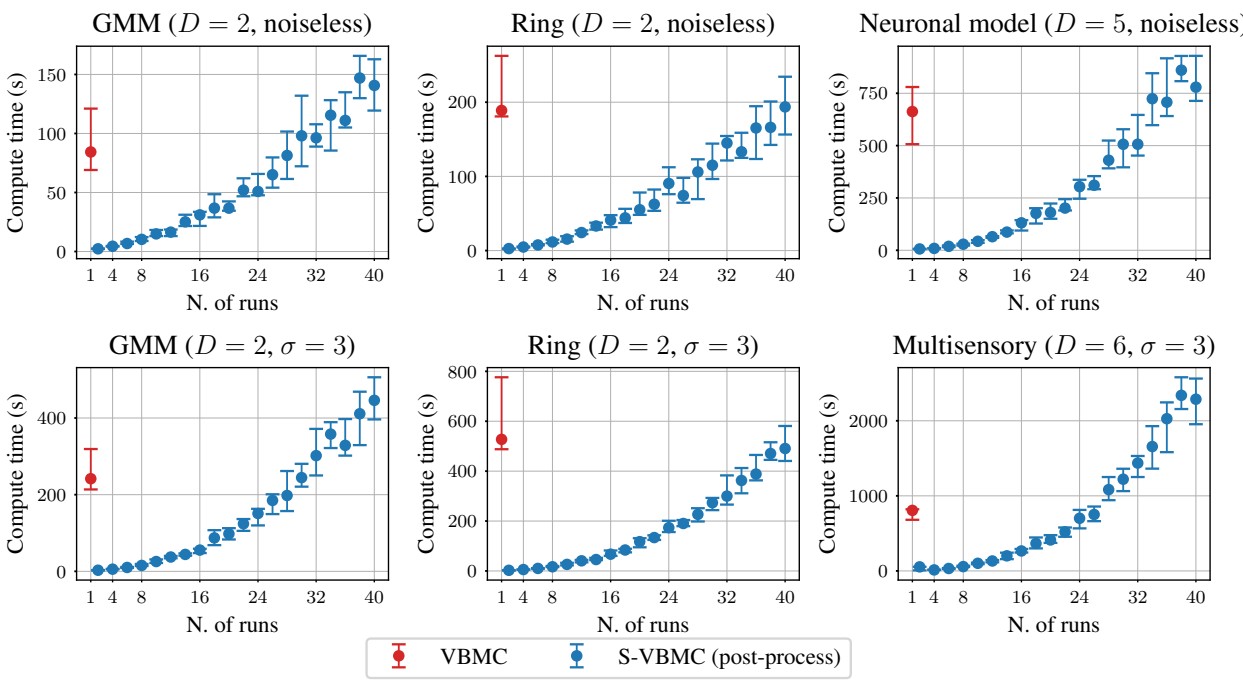

Figure 5: Compute time of a single VBMC run (red) and *post-processing time* only (*i.e.*, computational overhead) of S-VBMC (blue) plotted as a function of the number of VBMC runs stacked (median and 95% confidence interval, computed from 10000 bootstrap resamples). Each subplot represents a different benchmark problem. The values plotted here correspond to the actual computation times of the experiments described in Sections 4.4 and 4.5.

## 5 ELBO estimation bias

As briefly mentioned in Section 4.4, in our results we observe that, in noisy settings, while the ELBO keeps increasing as more VBMC runs are stacked, the $\Delta$LML error also increases, after an initial decrease (see the second column of Figures 3 and 4). This apparently odd result can be explained by the presence of a positive bias build-up in the estimated ELBO. This bias is visible in Figure 3 (b) and (d), first column, and Figure 4 (b), first column, in that the estimated ELBO from S-VBMC on these problems slightly "overshoots" the ground truth LML (the S-VBMC estimates, blue dots, end above the black horizontal line). As discussed in Section 2.2, the ELBO is *always* lower than the LML, with equality when the approximate posterior perfectly corresponds to the true posterior. However, in these results, the estimated ELBO grows larger than the ground-truth LML. As this cannot be true, there must be a positive bias in the ELBO estimate. As one can see in the aforementioned figures (first column), this bias builds up as more and more VBMC runs are stacked. What characterises these problems is the fact that they use a stochastic estimator for the likelihood (as opposed to exact likelihood evaluations used in the other problems, where the estimate does not overshoot).

While this bias surprisingly does not affect other posterior quality metrics, which keep improving (or plateau) with increasing $M$, it might constitute an issue when using $\text{ELBO}_{\text{stacked}}$ for model comparison. In this section, we provide a simplified model for how the bias might statistically arise in terms of the "winner's curse" and then provide an effective heuristic to counteract the bias.

### 5.1 Origin of bias

Here we analyse the ELBO overestimation observed in our results through a simplified example that illustrates one potential mechanism for this bias. In short, we suggest this occurs because all $I_{m,k}$ are noisy estimates of the true expected log-joint contributions, causing S-VBMC to overweigh the *most overestimated*

mixture components – an effect that increases with the number of components $M$. While other factors may contribute, this analysis provides insight into why the bias tends to increase with the number of merged VBMC runs.

For the sake of argument, consider $M$ VBMC runs that return *identical* posteriors $q_{\phi_1}(\boldsymbol{\theta}) = \ldots = q_{\phi_M}(\boldsymbol{\theta})$, each with a single component. The stacked posterior takes the form:

$$q_{\tilde{\phi}}(\boldsymbol{\theta}) = \sum_{m=1}^{M} \tilde{w}_m q_{\phi_m}(\boldsymbol{\theta}). \tag{23}$$

For each single-component posterior, the expected log-joint is approximated as

$$I_m = \mathbb{E}_{q_{\phi_m}}\left[\log p(\mathcal{D}|\boldsymbol{\theta})p(\boldsymbol{\theta})\right] \approx \mathbb{E}_{q_{\phi_m}}\left[f_m(\boldsymbol{\theta})\right] \tag{24}$$

where $f_m(\boldsymbol{\theta})$ is the surrogate log-joint from the $m$-th VBMC run. Since all posteriors share identical parameters, their entropies are equal:

$$\mathcal{H}\left[q_{\phi_1}(\boldsymbol{\theta})\right] = \mathcal{H}\left[q_{\phi_2}(\boldsymbol{\theta})\right] = \ldots = \mathcal{H}\left[q_{\phi_M}(\boldsymbol{\theta})\right]. \tag{25}$$

The stacked posterior is thus a mixture of identical components with different associated values $I_m$. The optimal mixture weights $\tilde{\mathbf{w}}$ depend solely on the noisy estimates of $I_m$:

$$\hat{I}_m = \mathbb{E}_{q_{\phi_m}}\left[f_m(\boldsymbol{\theta})\right] = \mathbb{E}_{q_{\phi_m}}\left[\log p(\mathcal{D}|\boldsymbol{\theta})p(\boldsymbol{\theta})\right] + \epsilon_m \tag{26}$$

where $\epsilon_m \sim \mathcal{N}(0, J_m)$ represents estimation noise with variance $J_m$. Since all posteriors are identical and derived from the same data and model, differences in expected log-joint estimates arise purely from noise deriving from the Gaussian process surrogates $f_m$.

Given that entropy remains constant under merging, in this scenario optimising $\text{ELBO}_{\text{stacked}}$ reduces to selecting the posterior with the highest expected log-joint estimate. If we denote $\hat{I}_{\max} = \max_m \hat{I}_m$, the optimal ELBO becomes

$$\text{ELBO}^*_{\text{stacked}} = \hat{I}_{\max} + \mathcal{H}\left[q_{\tilde{\phi}}(\boldsymbol{\theta})\right]. \tag{27}$$

Since the true expected log-joint is identical across posteriors, the optimisation selects the most overestimated value. The magnitude of this overestimation increases with both $M$ and the observation noise for $f_m$, introducing a positive bias in $\text{ELBO}^*_{\text{stacked}}$ that grows with the number of stacked runs and is more substantial for surrogates obtained from noisy log-likelihood observations.

While this simplified scenario does not capture the complexity of practical applications – where posteriors have multiple, non-overlapping components – it illustrates a fundamental issue: if we model each $\hat{I}_{m,k}$ as the sum of the true $I_{m,k}$ and noise, the merging process will favour overestimated components, biasing the final $\text{ELBO}_{\text{stacked}}$ estimate upward.

This hypothesis is substantiated by our results, as we only observe a noticeable bias in problems with noisy targets, where levels of noise in the VBMC estimation of $I_{m,k}$ are non-negligible (note that VBMC outputs an estimate of such noise, see Section 2).

## 5.2 Debiasing heuristic

We propose here a heuristic approach to counteract the bias in the ELBO estimate.

First, as a baseline we can estimate the per-component ground-truth expected log-joint $I_{m,k}$, and the full expected log-joint summed over components, using Monte Carlo sampling on the true log-joint instead of the VBMC estimates (see Eqs. 8 and 11). This estimate is unbiased and does not increase with $M$, as opposed to the one estimated by S-VBMC, as shown for all noisy benchmarks in Figure 6. However, this requires numerous additional evaluations of the model's log-likelihood, which is assumed to be expensive, so it is not in general a viable solution.

One potentially effective heuristic to counteract the bias – or at least prevent it from increasing with $M$ – would be to cap its value post-hoc to some reasonable quantity. This heuristic has the advantage of being straightforward and inexpensive to implement, and, crucially, of not involving any tweaks to the ELBO optimisation process, thus not requiring any additional hyperparameters.

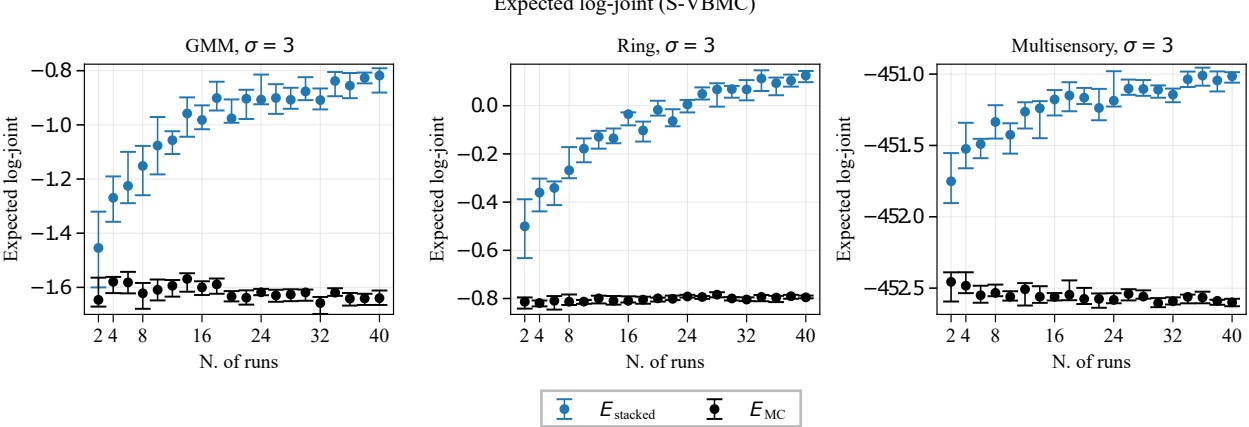

Figure 6: Expected log-joint as estimated by S-VBMC ($E_{\text{stacked}}$, blue) compared to that estimated via Monte Carlo sampling with numerous additional evaluations of the true log-joint ($E_{\text{MC}}$, black, which we use as the ground truth) for all our experiments with noisy targets. Dots represent the median value (of the 20 S-VBMC runs) and error bars 95% confidence intervals (computed from 10000 bootstrap resamples).

If we define the expected log-joint as estimated by S-VBMC as

$$E_{\text{stacked}} = \sum_{m=1}^{M} \sum_{k=1}^{K_m} \tilde{w}_{m,k} \hat{I}_{m,k}, \tag{28}$$

the 'capped' ELBO will be

$$\text{ELBO}_{\text{stacked}}^{(\text{capped})} = \widehat{E}_{\text{stacked}} + \mathcal{H}\left[ q_{\tilde{\phi}}(\boldsymbol{\theta}) \right], \tag{29}$$

where

$$\widehat{E}_{\text{stacked}} = \min\left( E_{\text{stacked}}, E_{\text{cap}} \right). \tag{30}$$

Note that, as mentioned in Section 3.1, each VBMC run performs its own parameter transformation $g_m(\cdot)$ during inference (Acerbi, 2020), which prevents meaningful comparisons of the expected log-joint and its components across runs. Throughout this section, we refer to and use the *corrected* VBMC estimates $\hat{I}_{m,k}$ as expressed in the common parameter space of $\boldsymbol{\theta}$. More details on this correction can be found in Appendix A.2. In what follows, we discuss two candidates for $E_{\text{cap}}$.

**Capping with median expected log-joint (run-wise).** As a possible candidate for $E_{\text{cap}}$, we considered the median of the VBMC estimates of the expected log-joint from individual runs. So, if

$$E_m = \sum_{k=1}^{K_m} w_{m,k} \hat{I}_{m,k} \tag{31}$$

is the VBMC estimate of the expected log-joint from the $m$-th individual run, then

$$E_{\text{median}} = \underset{1 \leq m \leq M}{\text{median}}(E_m). \tag{32}$$

**Capping with median expected log-joint (component-wise).** As the value of $E_{\text{median}}$ might be unstable and heavily dependent on individual VBMC runs (especially for low values of $M$), we also consider the median of the expected log-joints with respect to all individual components,

$$I_{\text{median}} = \underset{\substack{1 \le m \le M \\ 1 \le k \le K_m}}{\text{median}} (\hat{I}_{m,k}). \tag{33}$$

As each VBMC run typically has $K_m = 50$, even with $M = 2$ we would have a stacked posterior with 100 components, which should ensure more stability in the estimate.

**Debiasing results.** We applied both these corrections to all our experiments with noisy targets (described in Sections 4.4 and 4.5), with results shown in Figure 7. We observe that both solutions are effective in preventing the ELBO bias buildup for all three benchmark problems. The bias itself is still present for the ring and multisensory targets, but it remains roughly constant with increasing values of $M$, and in both cases is very limited. We also observe that using $I_{\text{median}}$ as $E_{\text{cap}}$ yields the most stable solution, with debiased ELBO values fluctuating less, and confidence intervals being considerably less wide. While this is true for all three scenarios, it is particularly evident in the multisensory benchmark, where the ELBO capped with $E_{\text{median}}$ tends to fluctuate wildly, sometimes leading to ELBO overestimation, and sometimes to ELBO underestimation. In contrast, capping with $I_{\text{median}}$ ensures a reliably contained bias ($< 0.5$). These results suggest that the latter method is more reliable across benchmarks and should therefore be preferred by practitioners.

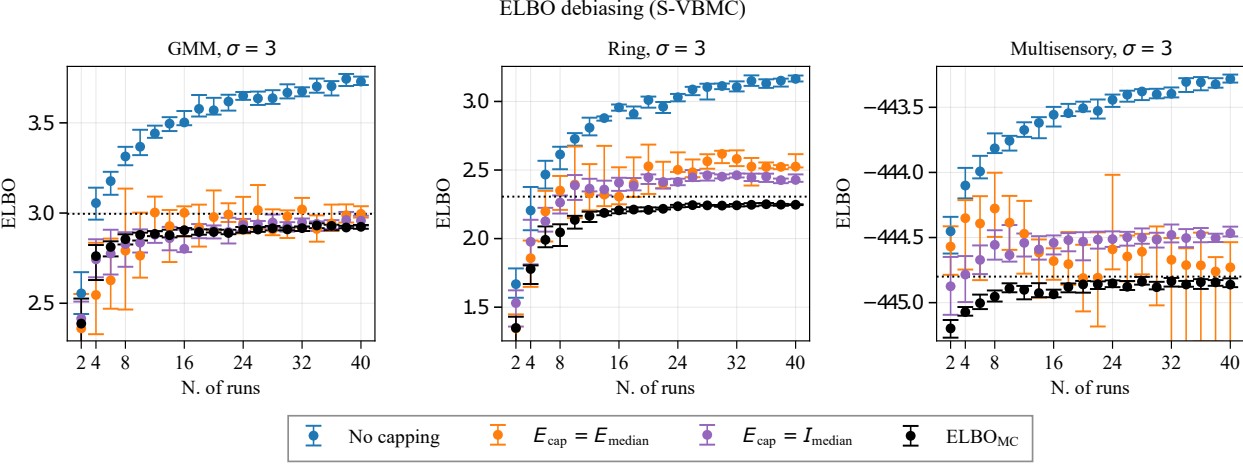

Figure 7: Effects of debiasing heuristics on noisy targets, with dots representing the median value and error bars 95% confidence intervals (computed from 10000 bootstrap resamples) across 20 S-VBMC runs. Each panel displays the uncorrected ELBO as output by S-VBMC (blue), the capped ELBO using $E_{\text{median}}$ (orange), the capped ELBO using $I_{\text{median}}$ (purple), and the ground-truth ELBO obtained via Monte Carlo $\text{ELBO}_{\text{MC}}$ (black), for distinct benchmark problems. In each panel, the black dotted line is the ground-truth log marginal likelihood.

## 6 Discussion

The core problem we tackled in this work is that certain target posteriors might have properties (*e.g.*, multimodality, long tails, long and narrow shapes) that pose significant challenges to global surrogate-based approaches to Bayesian inference like VBMC. VBMC in particular was developed to tackle problems with expensive likelihood functions, and thus relies on a limited target evaluation budget and an active sampling strategy (Acerbi, 2019), which, while effective in many cases (Acerbi, 2018; 2020), makes it vulnerable to leaving high-density probability regions unexplored (*e.g.*, missing a mode). Therefore, attempting to build

a single, global surrogate approximation of complex targets might be counter-productive. Our proposed solution relies on the idea that, as an alternative, we can rely on a collection of local surrogates to build a better, global posterior.

In this work, we introduced S-VBMC, a simple approach for merging independent VBMC runs in a principled way to yield a global posterior approximation. We tested its effectiveness on synthetic problems that we expected VBMC to have trouble with, such as targets with multiple modes and long, narrow shapes, as well as on two real-world problems from computational neuroscience. We also probed S-VBMC's robustness to noise, by adding Gaussian noise to log-likelihood evaluations for some of our benchmark targets. This approach mimics realistic scenarios where the log-likelihood itself is not available, but can be estimated via simulation (Wood, 2010; van Opheusden et al., 2020; Järvenpää et al., 2021; Acerbi, 2020). As expected, our results show that individual VBMC runs fail to yield good posterior approximations in our challenging problems. Conversely, S-VBMC is remarkably effective across benchmarks, with all metrics steadily improving as more VBMC runs are merged, with minor differences between noisy and noiseless settings. Importantly, we built S-VBMC to be robust to variability in the quality of individual VBMC posteriors. We initialise the weights $\tilde{\mathbf{w}}$ using each run's ELBO (Eq. 16), which immediately downweights low-ELBO runs, and the subsequent optimisation further reduces their influence. To verify this robustness, in all benchmarks we deliberately retained low-ELBO (but converged) runs and only excluded those flagged as non-converged by `pyvbmc` (Huggins et al., 2023) and poorly converged ones, with large uncertainty associated with the estimates $\hat{I}_k$ (see Section 4.1). Despite this, the stacked posterior still improved as more runs were combined, suggesting that low-ELBO VBMC posteriors are naturally assigned negligible weight.

S-VBMC inherits the limitations of VBMC and other surrogate-based inference methods, such as applicability to relatively low-dimensional target posteriors (up to $\approx 10$ dimensions; see Acerbi, 2018; 2020; Järvenpää et al., 2021). While this fundamental constraint of the Gaussian process surrogate remains, it is possible that, in moderately high dimensions ($D \approx 10 - 15$), diverse initialisations might allow different runs to capture complementary regions of the posterior, yielding incremental gains even when each surrogate is only locally accurate. However, we do not expect S-VBMC to overcome the core scaling issues of the surrogate itself, and a careful empirical study of higher-dimensional cases is left for future work. With that being said, S-VBMC effectively addresses some of VBMC's limitations, such as dealing with more complex posterior shapes and multimodality.

Our results show that S-VBMC represents a practical and effective approach for performing approximate Bayesian inference when the target log-likelihood is expensive or the posterior distribution exhibits specific features. In scenarios where the likelihood is fast to evaluate and noiseless, and the posterior largely unimodal, traditional inference methods such as Markov Chain Monte Carlo (MCMC) are likely to outperform S-VBMC and should remain the default choice. While methods like MCMC represent a gold standard in most cases, it is worth noting that standard MCMC algorithms can struggle to efficiently explore complex posteriors with multiple, well-separated modes, for which there is no off-the-shelf, easy solution. In contrast, S-VBMC's strategy of combining multiple independent local approximations, when paired with a diverse set of starting points, makes it better suited to explore challenging global structures. This suggests S-VBMC may be viable even in scenarios where the likelihood is not prohibitively expensive, but the posterior geometry is difficult for sequential samplers to navigate due to isolated modes.

Moreover, S-VBMC integrates seamlessly into the established best practices for its parent algorithm. For robustness and convergence diagnostics, performing several independent VBMC runs from different starting points is already recommended (Huggins et al., 2023). S-VBMC leverages this existing diagnostic workflow, providing a principled method to not only assess the inference landscape but to combine these runs into an improved, global posterior approximation at a small added computational cost. We intentionally designed S-VBMC as a simple post-processing step that requires no modification to the core VBMC algorithm. This simplicity is a key strength, as it preserves the *embarrassingly parallel* nature of running multiple independent inferences. While one could devise more sophisticated methods involving, for example, communication between runs to promote diversity and exploration, such approaches would sacrifice the ease of implementation and seamless integration that make S-VBMC a practical tool for practitioners. As our results show, this approach is most impactful for problems with features like multiple modes (GMM target), long and nar-

row shapes (ring target), or heavy tails (neuronal model, see Appendix A.5), where individual runs explore different facets of the posterior.

To investigate the practical convenience of our approach, we measured and reported the computational overhead of S-VBMC. Our results, combined with our previous analyses, suggest that our approach yields considerable improvements in terms of posterior quality with a small added cost in terms of compute time, assuming VBMC runs can be executed in parallel. In light of this, we recommend using $M = 10$ as a default choice, as it offers a strong accuracy–cost trade-off: most of the gains materialise by $M = 10$, which in our experiments added a modest $\approx 5 - 15\%$ overhead cost, assuming the VBMC runs are executed in parallel. As mentioned earlier, our method is, in principle, applicable for stacking mixture posteriors produced by any inference scheme, not strictly limited to VBMC. However, without a closed-form solution for the expected log-joints of the individual components of each run ($\{\mathbf{I}_m\}_{m=1}^{M}$, estimates of which are available in VBMC), Monte Carlo estimates are required, greatly inflating the number of necessary likelihood evaluations, and thus the computational cost. This wouldn't be the case for inexpensive likelihoods, but, as discussed above, VBMC would not necessarily be the primary recommended approach.

Finally, we discussed the main noticeable downside of S-VBMC, namely an observed bias in the ELBO estimation building up with increasing numbers of merged VBMC runs in problems with noisy targets, and proposed a simple heuristic (applicable in an inexpensive post-processing step) that mitigates this. Even though this bias buildup didn't seem to affect posterior quality in our experiments, it would constitute a problem if one were to use the (inflated) ELBO estimates for model comparison. Our debiasing approach reduces this bias to a smaller magnitude and prevents it from building up, so we encourage practitioners to adopt it (or some other debiasing technique) when using S-VBMC estimates of the ELBO for model comparison in problems with noisy likelihoods.

We should note that, although we discussed a plausible candidate source for the ELBO bias build-up in Section 5.1, this work does not contain a thorough investigation of this phenomenon and its causes. Some of the bias could come from the internals of the VBMC algorithm itself, which could explain the presence of a residual bias (see Section 5.2). Furthermore, our debiasing method simply consists of a post-processing heuristic, which is attractive for its simplicity and speed, but preserves the bias build-up mechanism during optimisation and inference. Precise identification of bias sources could allow their neutralisation at inference time, possibly leading the algorithm to shift its focus from overestimating the expected log-joint to optimising an unbiased stacked ELBO. Nonetheless, an interesting finding is that in our benchmark problems the bias phenomenon did not affect posterior approximation quality as measured by our metrics, which kept improving steadily and approaching ground-truth posteriors while stacking more VBMC runs.

## 7 Conclusion

In this work, we introduced S-VBMC, a simple, novel approach for stacking variational posteriors generated by separate, possibly parallel, VBMC runs, and tested it on a set of challenging targets. We further suggested an effective and inexpensive method to address one main drawback of our approach (*i.e.*, the ELBO bias build-up). Our results, both in terms of performance and compute time, show its practical convenience for VBMC users, especially when tackling particularly challenging inference problems.

**Acknowledgments**

This work was supported by Research Council of Finland (grants 358980 and 356498). The authors wish to thank the Finnish Computing Competence Infrastructure (FCCI) for supporting this project with computational and data storage resources. The authors also acknowledge the research environment provided by ELLIS Institute Finland.

The model referred to as the "neuronal model" in the main text and appendices was developed by Dániel Terbe, Balázs Szabó and Szabolcs Káli (HUN-REN Institute of Experimental Medicine, Budapest, Hungary), using data collected by Miklós Szoboszlay in Zoltán Nusser's laboratory (HUN-REN Institute of Experimental Medicine, Budapest, Hungary). The authors thank these researchers for making their data and model available for the work described in this paper.

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

# A  Appendix

This appendix provides additional details and analyses to complement the main text, included in the following sections:

- An overview of Variational Bayesian Monte Carlo, A.1

- A description of how S-VBMC handles VBMC's parameter transformations, A.2

- Additional experiments, A.3

- Full experimental results, A.4

- Example posterior visualisations, A.5

## A.1  An overview of Variational Bayesian Monte Carlo

In this appendix we briefly describe Variational Bayesian Monte Carlo (VBMC). This is a simple overview of the various components of the algorithm, and a full in-depth description of these is beyond the scope of this appendix. For further details, see Acerbi (2018; 2020).

As mentioned in Section 2.3, VBMC addresses the problem of expensive likelihoods with black-box properties by using a surrogate for the log-joint (see Eq. 5). Like many other surrogate-based approaches (Garnett, 2023), VBMC uses a Gaussian process (GP) to approximate its expensive target (Rasmussen & Williams, 2006). GPs are stochastic processes such that any finite collection $f(\mathbf{x}_1), \ldots, f(\mathbf{x}_n)$ follows a multivariate normal distribution. A GP is fully specified by a mean function

$$m(\mathbf{x}) = \mathbb{E}[f(\mathbf{x})], \tag{A.1}$$

a covariance function (also called a kernel)

$$\kappa(\mathbf{x}, \mathbf{x}') = \text{cov}\left[f(\mathbf{x}), f(\mathbf{x}')\right], \tag{A.2}$$

and an observation noise model or likelihood. In VBMC, as in most surrogate modelling approaches, the likelihood is assumed to be Gaussian, which affords closed-form GP posterior computations. Specifically, given a training set $(\mathbf{X}, \mathbf{y}, \mathbf{S})$ with $\mathbf{X}$ being the observed input locations, $\mathbf{y}$ the corresponding observed function values, and $\mathbf{S}$ a diagonal covariance matrix representing observation noise, the posterior mean and covariance functions for a test input location $\tilde{\mathbf{x}}$ are, respectively,

$$\mu_p(\tilde{\mathbf{x}}) = \kappa(\tilde{\mathbf{x}}, \mathbf{X})(\kappa(\mathbf{X}, \mathbf{X}) + \mathbf{S})^{-1}(\mathbf{y} - m(\mathbf{X})) + m(\tilde{\mathbf{x}}) \tag{A.3}$$

and

$$\kappa_p(\tilde{\mathbf{x}}, \tilde{\mathbf{x}}) = \kappa(\tilde{\mathbf{x}}, \tilde{\mathbf{x}}) - \kappa(\tilde{\mathbf{x}}, \mathbf{X})(\kappa(\mathbf{X}, \mathbf{X}) + \mathbf{S})^{-1}\kappa(\mathbf{X}, \tilde{\mathbf{x}}). \tag{A.4}$$

For further details on GPs and their use in machine learning, see Rasmussen & Williams (2006).

Crucially, a GP surrogate does not yield a usable posterior approximation, as the integral of Eq. 1 (Bayes' rule) remains intractable. Even if the integral can be solved, it does not yield a *usable* approximation of the posterior, such as the ability to draw samples from it. To address this point, VBMC makes use of Bayesian quadrature, a method for obtaining Bayesian estimates of intractable integrals (O'Hagan, 1991; Ghahramani & Rasmussen, 2002). Given an integral

$$\mathcal{J} = \int f(\mathbf{x})\pi(\mathbf{x})d\mathbf{x} \tag{A.5}$$

where $f$ is the target function and $\pi$ is a known probability distribution, if a GP prior is specified for $f$, the integral $\mathcal{J}$ is a Gaussian random variable with posterior mean

$$\mathbb{E}_f[\mathcal{J}] = \int \mu_p(\mathbf{x})\pi(\mathbf{x})d\mathbf{x} \tag{A.6}$$

and variance

$$\mathbb{V}_f[\mathcal{J}] = \int \int \kappa_p(\mathbf{x}, \mathbf{x}')\pi(\mathbf{x})\pi(\mathbf{x}')d\mathbf{x}d\mathbf{x}'. \tag{A.7}$$

If $f$ has a Gaussian kernel and $\pi(\mathbf{x})$ is a mixture of Gaussians – which is the case for the VBMC approximate posterior –, both integrals have closed-form solutions.

As mentioned in Section 2, VBMC leverages Bayesian quadrature to estimate the ELBO by building a surrogate model of the log-joint $f(\boldsymbol{\theta}) \approx \log p(\boldsymbol{\theta})p(\mathcal{D}|\boldsymbol{\theta})$. In particular, the GP model $f$ uses an *exponentiated quadratic* (more commonly known, if incorrectly, as squared exponential) kernel and a *negative quadratic* mean function (Acerbi, 2018; 2019). The latter ensures integrability of the function and is equivalent to an inductive bias towards Gaussian posteriors, but note that it does *not* limit the modelled target to be a Gaussian, as the GP can model arbitrary deviations from the mean function.

Therefore, putting everything together, the posterior mean of the surrogate ELBO can be calculated as

$$\mathbb{E}_f[\text{ELBO}(\boldsymbol{\phi})] = \mathbb{E}_f[\mathbb{E}_{\boldsymbol{\phi}}[f(\boldsymbol{\theta})]] + \mathcal{H}[q_{\boldsymbol{\phi}}(\boldsymbol{\theta})] \tag{A.8}$$

where $\mathbb{E}_f[\mathbb{E}_{\boldsymbol{\phi}}[f(\boldsymbol{\theta})]]$ is the posterior mean of the GP surrogate of the log-joint (*i.e.*, the expected value of the expected log-joint). Here the expected log-joint takes the form

$$\mathbb{E}_{\boldsymbol{\phi}}[f(\boldsymbol{\theta})] = \int f(\boldsymbol{\theta})q_{\boldsymbol{\phi}}(\boldsymbol{\theta})d\boldsymbol{\theta} \tag{A.9}$$

which, since $q_{\boldsymbol{\phi}}(\boldsymbol{\theta})$ is a mixture of Gaussians, affords closed-form solutions for its posterior mean and variance, as well as its gradients.

To build a good surrogate approximation of the log-joint, a number of likelihood evaluations are needed to train the GP. Assuming the likelihood is expensive to compute, it is desirable to keep this number relatively contained. VBMC tackles this through an iterative process. In each iteration, VBMC selects the next points to evaluate by optimising an acquisition function that trades off exploration of new posterior regions and exploitation of high-density regions (see Acerbi, 2019; 2020). It then uses the resulting log-joint values to refine the GP surrogate, which is in turn used to refine the variational posterior via Bayesian quadrature. The VBMC algorithm begins with only two mixture components in a *warm-up* stage used to construct an initial surrogate model (see Acerbi, 2018). After the initial warm-up iterations are concluded, VBMC starts adding new components to the mixture to refine the posterior approximation. This process of acquiring new points, using them to improve the surrogate model, and then refining the variational approximation (possibly increasing the number of components), is repeated until a convergence criterion is met or until the likelihood evaluation budget is exceeded.

## A.2 Change-of-variables corrections in S-VBMC

**Problem setting.** Due to the *variational whitening* feature introduced in Acerbi (2020), each VBMC run operates in its own transformed parameter space, whereas in Acerbi (2018) all VBMC runs shared the same transformed parameter space (a fixed transform for bounded variables). An interested reader should refer to Acerbi (2020) for more information about variational whitening, and an in-depth discussion of this is beyond the scope of this appendix.

In the context of the $m$-th VBMC run, we call $g_m(\cdot)$ the function determining the parameter transformation, $g_m(\boldsymbol{\theta})$ the transformed parameters, and $\boldsymbol{\psi}_m$ the parameters of the variational posterior expressed in the transformed space, $q_{\boldsymbol{\psi}_m}(g_m(\boldsymbol{\theta}))$.

Crucially, VBMC returns approximate posteriors in the transformed space, and, since each run is executed independently, and has its own transformation $g_m(\cdot)$, the densities obtained with the different approximate posterior parameters $q_{\boldsymbol{\psi}_m}(g_m(\boldsymbol{\theta}))$ are *not directly comparable*. To calculate the stacked ELBO, we need to operate in the common (original) parameter space, in which the parameters $\boldsymbol{\theta}$ are expressed.

Concretely, this means applying appropriate corrections to the densities used to evaluate the stacked ELBO. In the following, we provide a brief introduction to such corrections and discuss how they can be applied to compute the entropy and the expected log-joint (*i.e.*, the two terms of the stacked ELBO, see Eq. 11) in the common parameter space. Importantly, these corrections do not depend on the mixture weights of the stacked posterior, preserving the differentiability of our objective function with respect to $\tilde{\mathbf{w}}$.

**The Jacobian correction.** Parameter transformations can be handled via the *Jacobian correction*. Given a random variable $\boldsymbol{x}$, its probability $p_{\boldsymbol{x}}(\boldsymbol{x})$ and a transformation $\boldsymbol{y} = T(\boldsymbol{x})$, we have

$$p_{\boldsymbol{x}}(\boldsymbol{x}) = p_{\boldsymbol{y}}(T(\boldsymbol{x})) \left| \det \frac{\partial T(\boldsymbol{x})}{\partial \boldsymbol{x}} \right| \tag{A.10}$$

where the correction is applied with the absolute value of the determinant of the Jacobian of $T(\boldsymbol{x})$. Conveniently, the `pyvbmc` software (Huggins et al., 2023) provides a function to evaluate the (log) absolute value of the determinant of the Jacobian of the *inverse* transformation (evaluated at $g_m(\boldsymbol{\theta})$), which, in this toy example, would mean

$$p_{\boldsymbol{x}}(\boldsymbol{x}) = \frac{p_{\boldsymbol{y}}(T(\boldsymbol{x}))}{\left| \det \frac{\partial T^{-1}(T(\boldsymbol{x}))}{\partial T(\boldsymbol{x})} \right|}, \tag{A.11}$$

To tidy our notation, we will call our corrections term for the $m$-th transformation

$$J_m(g_m(\boldsymbol{\theta})) \equiv \left| \det \frac{\partial g_m^{-1}(g_m(\boldsymbol{\theta}))}{\partial g_m(\boldsymbol{\theta})} \right|. \tag{A.12}$$

Bringing this back to our case, the variational posterior of the $m$-th VBMC run can be written as

$$q_{\boldsymbol{\phi}_m}(\boldsymbol{\theta}) = \sum_{k=1}^{K_m} w_{m,k} \, q_{k,\boldsymbol{\psi}_m}(g_m(\boldsymbol{\theta})) J_m^{-1}(g_m(\boldsymbol{\theta})), \tag{A.13}$$

and the stacked posterior as

$$q_{\tilde{\boldsymbol{\phi}}}(\boldsymbol{\theta}) = \sum_{m=1}^{M} \sum_{k=1}^{K_m} \tilde{w}_{m,k} \, q_{k,\boldsymbol{\psi}_m}(g_m(\boldsymbol{\theta})) J_m^{-1}(g_m(\boldsymbol{\theta})). \tag{A.14}$$

In line with the notation used in the main text, $\boldsymbol{\phi}_m$ and $\tilde{\boldsymbol{\phi}}$ are the parameters of the $m$-th VBMC posterior and of the stacked posterior, respectively, expressed in the common parameter space. With the exception of $\mathbf{w}_m$ and $\tilde{\mathbf{w}}$ (which are not affected by the transformation), these parameters are unknown, but, as we will show in the following paragraphs, they are not needed to estimate the stacked ELBO.

**The corrected entropy.** One term of the stacked ELBO is the entropy

$$\mathcal{H}[q_{\tilde{\boldsymbol{\phi}}}] = -\mathbb{E}_{q_{\tilde{\boldsymbol{\phi}}}}[\log q_{\tilde{\boldsymbol{\phi}}}(\boldsymbol{\theta})], \tag{A.15}$$

for which no closed-form solution is available. We estimate it via Monte Carlo as in Eq. 12 of the main text, but crucially *evaluate all component densities in the original space* using the correction shown in Eqs. A.13 and A.14. Concretely, for each component $q_{k,\boldsymbol{\psi}_m}$ we draw $S$ samples in the $m$-th transformed space, $\left\{\mathbf{z}_{m,k}^{(s)} \sim q_{k,\boldsymbol{\psi}_m}\right\}_{s=1}^{S}$, and map them to the original space, $\mathbf{x}_{m,k}^{(s)} = g_m^{-1}(\mathbf{z}_{m,k}^{(s)})$. Then, for every sample $\mathbf{x}_{m,k}^{(s)}$ and for every component $q_{k',\boldsymbol{\phi}_{m'}}$, we compute the per-component log-density in the original space via

$$\log q_{k',\boldsymbol{\phi}_{m'}}(\mathbf{x}_{m,k}^{(s)}) = \log q_{k',\boldsymbol{\psi}_{m'}}(g_{m'}(\mathbf{x}_{m,k}^{(s)})) - \log J_{m'}(g_{m'}(\mathbf{x}_{m,k}^{(s)})), \tag{A.16}$$

and then aggregate

$$\log q_{\tilde{\boldsymbol{\phi}}}(\mathbf{x}_{m,k}^{(s)}) = \log \sum_{m'=1}^{M} \sum_{k'=1}^{K_{m'}} \tilde{w}_{m',k'} q_{k',\boldsymbol{\phi}_{m'}}(\mathbf{x}_{m,k}^{(s)}) \tag{A.17}$$

via log-sum-exp. Then these values can be plugged into Eq. 12 of the main text to estimate the entropy. Importantly, the transformations do not depend on the mixture weights, so the entropy remains differentiable with respect to $\tilde{\mathbf{w}}$.

**The corrected expected log-joint.** Let $\hat{L}_{m,k}$ be the VBMC estimate (computed in the run's transformed coordinates) of the component-wise expected log-joint,

$$\hat{L}_{m,k} \approx \mathbb{E}_{q_{k,\boldsymbol{\psi}_m}}\left[\log p_m(\mathcal{D}, g_m(\boldsymbol{\theta}))\right], \tag{A.18}$$

where $p_m(\mathcal{D}, g_m(\boldsymbol{\theta}))$ is the log-joint reparametrised with the $m$-th transform. To express this expectation in the *common* parameter space we apply the correction determined by $g_m(\boldsymbol{\theta})$

$$\hat{I}_{m,k} = \hat{L}_{m,k} - \mathbb{E}_{q_{k,\boldsymbol{\psi}_m}}\left[\log J_m(g_m(\boldsymbol{\theta}))\right] \approx \mathbb{E}_{q_{k,\boldsymbol{\phi}_m}}\left[\log p(\mathcal{D}, \boldsymbol{\theta})\right], \tag{A.19}$$

In practice we estimate the (per–component) Jacobian term in Eq. A.19 using the same samples $\left\{\mathbf{z}_{m,k}^{(s)} \sim q_{k,\boldsymbol{\psi}_m}\right\}_{s=1}^{S}$ employed for the entropy and setting

$$\mathbb{E}_{q_{k,\boldsymbol{\psi}_m}}\left[\log J_m(g_m(\boldsymbol{\theta}))\right] \approx \frac{1}{S} \sum_{s=1}^{S} \log J_m\left(\mathbf{z}_{m,k}^{(s)}\right). \tag{A.20}$$

After applying these corrections, and those to the entropy described above, we can calculate the corrected stacked ELBO as in Eq. 11 of the main text.

Importantly, the $\hat{I}_{m,k}$ described here (*i.e.*, the corrected ones) are the values we use in Section 5.2 for debiasing the stacked ELBO in noisy settings.

### A.3  Additional experiments

### A.3.1  S-VBMC variant

To probe the benefits of optimising the ELBO with respect to the weights of the individual components, we performed additional experiments comparing the version of S-VBMC presented in the main text ("all-weights") with a S-VBMC variant where we only reweigh the weights of each individual VBMC posterior ("posterior-only"). Specifically, we considered a stacked posterior written as:

$$q_{\tilde{\phi}}(\boldsymbol{\theta}) = \sum_{m=1}^{M} \tilde{\omega}_m q_{\phi_m}(\boldsymbol{\theta}). \tag{A.21}$$

For this "posterior-only" variant, we optimised the global ELBO with respect to the weights $\tilde{\omega}_m$ assigned to each posterior. This is similar to the naive stacking approach seen in the main paper (Eq. 18), with the difference that the posterior weights are now optimised. We ran this method for all the benchmark problems described in Sections 4.4 and 4.5, using the same bootstrapping procedure described in Section 4.1.

The results of this comparison are reported in Figures A.1 and A.2, and in further detail in Appendix A.4.2. We observe that optimising with respect to $\tilde{\boldsymbol{\omega}}$ ("posterior-only") performs well, with both MMTV and GsKL metrics steadily improving with increased numbers of stacked posteriors. In fact, for most problems, "posterior-only" S-VBMC performs comparably to the "all-weights" variant presented in the main paper, which optimises all components weights $\tilde{\mathbf{w}}$. Still, the "all-weights" variant performs slightly better in the GsKL metric and in some challenging scenarios (*e.g.*, the multisensory model), so it remains our base recommendation, paired with the debiasing approach described in Section 5.

### A.3.2  Additional runtime analyses

Here we report the total runtime cost (in seconds) of BBVI for all our benchmark problems compared to that of running VBMC 40 times and stacking the resulting posteriors with S-VBMC. We consider this particular number of runs because the BBVI target evaluation budget was set to match that of S-VBMC with $M = 40$. For both, we report the median and 95% confidence interval, computed from 10000 bootstrap resamples. For S-VBMC, each resample consisted of 40 VBMC runs and one S-VBMC run, then the S-VBMC runtime was added to that of the VBMC run with the *highest* runtime. This follows from the assumption that VBMC is run 40 times in parallel, and S-VBMC can be launched the moment the *last* VBMC run has converged.

It is important to note that, as is common in the surrogate-based literature (Acerbi, 2018; Wang & Li, 2018; Acerbi, 2020; Järvenpää et al., 2021; El Gammal et al., 2023; Järvenpää & Corander, 2024), in this work, we demonstrated the efficacy of our method on several problems where function evaluations are not computationally expensive, as a full benchmark with multiple expensive models is highly impractical. Therefore, wall-clock time needs to be interpreted carefully as a metric when comparing methods with different likelihood evaluation costs.

We can directly compare VBMC and S-VBMC, as we did in Section 4.6, because by construction they use the same backbone method and have the same evaluation costs (S-VBMC adds a small post-processing cost, which, crucially, does not depend on the cost of likelihood evaluation). Conversely, comparisons to non-VBMC methods become highly problem-dependent. The typical solution would consist of matching the number of function evaluations (as we did, see Section 4.2), for which non-surrogate-based baselines would be at a significant disadvantage, as demonstrated in previous work (Acerbi, 2018; 2020).

These considerations are crucial to interpret these results, displayed in Table A.1. As expected, BBVI is much faster than S-VBMC on problems with fast likelihood evaluation, but as soon as the likelihood becomes more expensive ($\approx 0.7$ seconds per evaluation for the neuronal model) the cost of non-VBMC methods increases dramatically, illustrating the kind of scenarios VBMC was developed to solve in the first place.

Finally, it is worth noting that, as shown in Figures 3 and 4 and Tables A.3 and A.4, BBVI performs substantially worse than S-VBMC across our examples, particularly in our real-world problems. Therefore, even where there may be runtime advantages (with the caveats discussed above), these come at a considerable cost in terms of posterior quality.

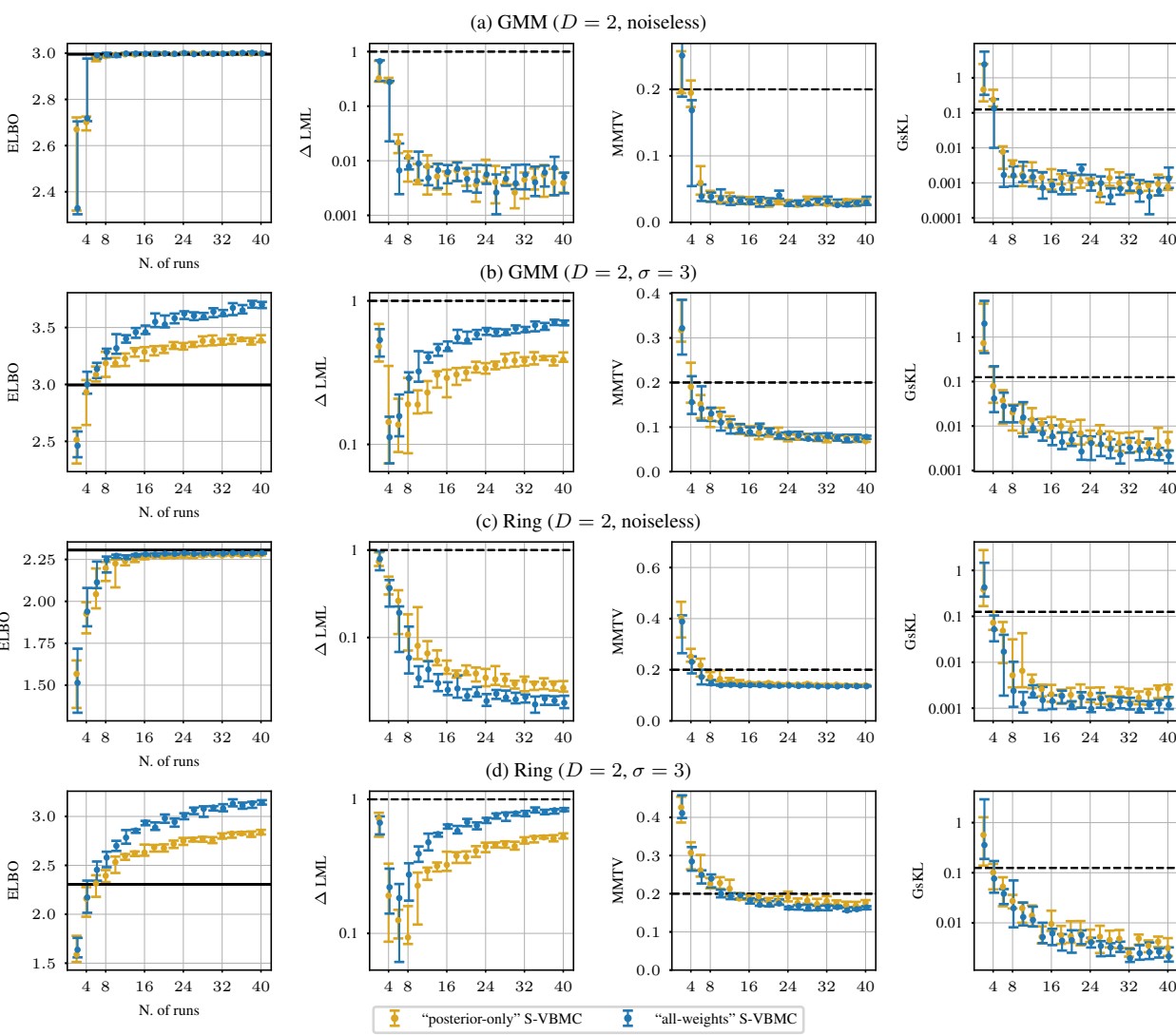

Figure A.1: Performance comparison between the two versions of S-VBMC ("all-weights" and "posterior-only") on synthetic problems. Metrics are plotted as a function of the number of VBMC runs stacked (median and 95% confidence interval, computed from 10000 bootstrap resamples) for S-VBMC when the ELBO is optimised with respect to "all-weights" (blue) and "posterior-only" weights (yellow). The black horizontal line in the ELBO panels represents the ground-truth LML, while the dashed lines on ΔLML, MMTV, and GsKL denote desirable thresholds for each metric (good performance is below the threshold; see Section 4.3)

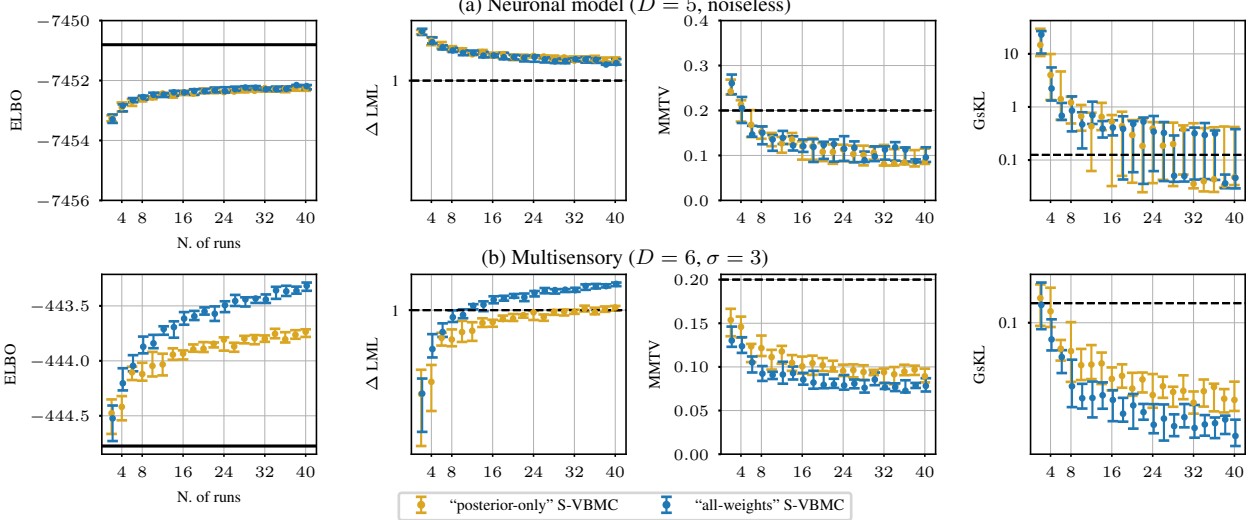

Figure A.2: Performance comparison between the two versions of S-VBMC ("all-weights" and "posterior-only") on real-world problems. Metrics are plotted as a function of the number of VBMC runs stacked (median and 95% confidence interval, computed from 10000 bootstrap resamples) for S-VBMC when the ELBO is optimised with respect to "all-weights" (blue) and "posterior-only" weights (yellow). The black horizontal line in the ELBO panels represents the ground-truth LML, while the dashed lines on ΔLML, MMTV, and GsKL denote desirable thresholds for each metric (good performance is below the threshold; see Section 4.3)

Table A.1: BBVI runtime (in seconds) compared to that of 40 (parallel) VBMC runs and their subsequent stacking with S-VBMC. Values show median with 95% confidence interval in brackets. Bold entries indicate the best median performance (*i.e.*, lowest compute time).

| | Algorithm | |
| --- | --- | --- |
| **Benchmark** | **BBVI runtime (s)** | **VBMC + S-VBMC runtime (s)** |
| GMM (noiseless) | **9.9** [9.4,10.3] | 458.3 [428.3,510.8] |
| GMM ($\sigma = 3$) | **12.8** [12.2,13.7] | 857.8 [759.0,954.8] |
| Ring (noiseless) | **12.2** [11.7,12.6] | 559.3 [516.9,794.2] |
| Ring ($\sigma = 3$) | **14.0** [13.4,15.0] | 1269.9 [1206.0,1557.4] |
| Neuronal model (noiseless) | 8497.0 [8411.2,8617.8] | **1561.8** [1445.1,1900.1] |
| Multisensory model ($\sigma = 3$) | **24.5** [21.2,27.3] | 3149.5 [2616.3,3525.1] |

### A.4    Full experimental results

### A.4.1    Filtering procedure

Here we briefly present the results of our filtering procedure, described in Section 4.1. As shown in Table A.2, VBMC had considerable trouble when run on the neuronal model, with over half the runs failing to converge (as assessed by the `pyvbmc` software, Huggins et al., 2023), suggesting a complex, non-trivial posterior structure which is reflected in the poor performance of other inference methods (see Figure 4 and Table A.4). Convergence issues were also found with the noisy Ring target, although to a lesser extent. Once non-converged runs were discarded, our second filtering criterion (*i.e.*, excluding poorly converged runs with excessive uncertainty associated with the $\hat{I}_k$ estimates) led to considerably fewer exclusions overall, with only the Ring target being somewhat affected (8% and 13% of runs discarded in noiseless and noisy settings, respectively).

All our VBMC runs were indexed, and, for our experiments, we used the 100 filtered runs with the lowest indices.

Table A.2: Result of our filtering procedures. This table shows the total number of VBMC runs we performed ("Total"), those that did not converge ("Non-converged") and converged poorly ("Poorly converged") out of the total, and the ones that passed both filtering criteria ("Remaining").

| Benchmark | VBMC runs | | | |
| --- | --- | --- | --- | --- |
| | Total | Non-converged | Poorly converged | Remaining |
| GMM (noiseless) | 120 | 2 | 3 | 115 |
| GMM ($\sigma = 3$) | 150 | 0 | 5 | 145 |
| Ring (noiseless) | 120 | 3 | 9 | 108 |
| Ring ($\sigma = 3$) | 149 | 34 | 15 | 100 |
| Neuronal model (noiseless) | 300 | 159 | 1 | 140 |
| Multisensory ($\sigma = 3$) | 150 | 0 | 1 | 149 |

### A.4.2    Posterior metrics

We present a comprehensive comparison of S-VBMC against VBMC, NS and BBVI in Tables A.3 and A.4, complementing the visualisations in Figures 3, 4, A.1 and A.2. We consider both the version of S-VBMC described in the main text (where the ELBO is optimised with respect to the component weights $\tilde{\mathbf{w}}$, "all-weights"), and the one described in Appendix A.3.1 (where the ELBO is optimised with respect to the posterior weights $\tilde{\boldsymbol{\omega}}$, "posterior-only").

For both synthetic problems (Table A.3) and real-world problems (Table A.4), S-VBMC generally demonstrates consistently improved posterior approximation metrics compared to the baselines. However, we observe an increase in $\Delta$LML error with larger numbers of stacked runs in problems with noisy targets. This increase likely stems from the accumulation of ELBO estimation bias, a phenomenon analysed in detail in Section 5.

Table A.3: Comparison of S-VBMC, VBMC, and BBVI performance on synthetic benchmark problems. Values show median with 95% confidence intervals (computed from 10000 bootstrap resamples) in brackets. Bold entries indicate best median performance; multiple entries are bolded when confidence intervals overlap with the best median. For compactness, we label the S-VBMC version described in the main text "w.r.t. $\tilde{\mathbf{w}}$", (indicating that the ELBO is optimised with respect to "all-weights" $\tilde{\mathbf{w}}$), and the version described in Appendix A.3.1 "w.r.t. $\tilde{\boldsymbol{\omega}}$", (indicating that the ELBO is optimised with respect to $\tilde{\boldsymbol{\omega}}$, or "posterior-only").

| | Benchmarks | | | | | |
| | GMM | | | Ring | | |
| Algorithm | $\Delta$LML | MMTV | GsKL | $\Delta$LML | MMTV | GsKL |
|---|---|---|---|---|---|---|
| | Noiseless | | | | | |
| BBVI, MoG ($K=50$) | 0.059 [0.028,0.075] | **0.059** [0.035,0.08] | **0.0083** [0.0011,0.019] | 8 [6.8,9.6] | 0.51 [0.48,0.53] | 0.72 [0.66,1.2] |
| BBVI, MoG ($K=500$) | 0.053 [0.029,0.11] | 0.052 [0.043,0.07] | 0.0087 [0.0025,0.013] | 8.3 [6.9,12] | 0.47 [0.45,0.49] | 0.67 [0.55,0.81] |
| VBMC | 1.4 [0.7,1.4] | 0.54 [0.39,0.55] | 13 [7.6,14] | 1.2 [1.2,1.3] | 0.53 [0.51,0.56] | 9.4 [7.2,14] |
| NS (10 runs) | 0.091 [0.07,0.12] | 0.15 [0.12,0.17] | 0.054 [0.034,0.075] | 0.16 [0.09,0.24] | 0.19 [0.18,0.22] | 0.04 [0.021,0.091] |
| NS (20 runs) | 0.047 [0.037,0.062] | 0.11 [0.089,0.12] | 0.027 [0.018,0.032] | 0.11 [0.073,0.13] | 0.18 [0.16,0.18] | 0.028 [0.018,0.049] |
| S-VBMC (w.r.t. $\tilde{\boldsymbol{\omega}}$, 10 runs) | **0.0042** [0.0037,0.0087] | **0.032** [0.031,0.04] | **0.0017** [0.00081,0.003] | 0.08 [0.057,0.22] | 0.16 [0.15,0.2] | 0.0065 [0.0029,0.043] |
| S-VBMC (w.r.t. $\tilde{\boldsymbol{\omega}}$, 20 runs) | **0.0059** [0.0035,0.0074] | **0.031** [0.024,0.035] | **0.0011** [0.00064,0.0016] | 0.04 [0.037,0.048] | **0.15** [0.14,0.15] | **0.002** [0.0013,0.0028] |
| S-VBMC (w.r.t. $\tilde{\mathbf{w}}$, 10 runs) | **0.0089** [0.0043,0.015] | **0.036** [0.028,0.05] | **0.0015** [0.0011,0.004] | 0.034 [0.027,0.047] | **0.14** [0.14,0.14] | **0.0013** [0.00081,0.0023] |
| S-VBMC (w.r.t. $\tilde{\mathbf{w}}$, 20 runs) | **0.0046** [0.0028,0.0072] | **0.031** [0.026,0.036] | **0.0013** [0.00047,0.0019] | **0.022** [0.019,0.026] | **0.14** [0.13,0.14] | **0.0011** [0.00096,0.0014] |
| | Noisy ($\sigma=3$) | | | | | |
| BBVI, MoG ($K=50$) | 0.23 [0.11,0.43] | **0.13** [0.092,0.18] | 0.03 [0.01,0.12] | 4.3 [3.3,4.7] | 0.51 [0.47,0.54] | 1.1 [0.65,1.7] |
| BBVI, MoG ($K=500$) | 0.27 [0.076,0.45] | **0.1** [0.094,0.13] | 0.019 [0.011,0.034] | 4.7 [4,5.5] | 0.93 [0.91,0.94] | 48 [28,49] |
| VBMC | 0.98 [0.78,1.1] | 0.44 [0.43,0.47] | 9.7 [8.5,11] | 1.3 [1.1,1.5] | 0.62 [0.57,0.65] | 38 [24,95] |
| NS (10 runs) | **0.11** [0.066,0.19] | 0.17 [0.14,0.18] | 0.066 [0.029,0.12] | **0.082** [0.066,0.12] | 0.23 [0.21,0.26] | 0.056 [0.033,0.091] |
| NS (20 runs) | **0.056** [0.046,0.082] | **0.1** [0.09,0.12] | 0.017 [0.011,0.026] | 0.19 [0.16,0.22] | **0.18** [0.18,0.2] | 0.023 [0.017,0.03] |
| S-VBMC (w.r.t. $\tilde{\boldsymbol{\omega}}$, 10 runs) | 0.19 [0.16,0.24] | 0.13 [0.11,0.14] | **0.012** [0.0069,0.031] | **0.23** [0.12,0.28] | 0.23 [0.21,0.24] | 0.02 [0.01,0.026] |
| S-VBMC (w.r.t. $\tilde{\boldsymbol{\omega}}$, 20 runs) | 0.32 [0.28,0.34] | **0.089** [0.078,0.098] | **0.0082** [0.004,0.013] | 0.37 [0.34,0.41] | **0.18** [0.17,0.19] | **0.0054** [0.004,0.011] |
| S-VBMC (w.r.t. $\tilde{\mathbf{w}}$, 10 runs) | 0.32 [0.27,0.45] | **0.11** [0.092,0.13] | **0.016** [0.0058,0.034] | 0.39 [0.34,0.45] | 0.2 [0.19,0.21] | 0.013 [0.0089,0.025] |
| S-VBMC (w.r.t. $\tilde{\mathbf{w}}$, 20 runs) | 0.53 [0.51,0.61] | **0.09** [0.084,0.097] | **0.0049** [0.0036,0.0072] | 0.68 [0.63,0.71] | **0.17** [0.17,0.18] | **0.0045** [0.0025,0.0071] |

Table A.4: Comparison of S-VBMC, VBMC, and BBVI performance on neuronal and multisensory causal inference models. Bold entries indicate best median performance; multiple entries are bolded when confidence intervals overlap with the best median. See the caption of Table A.3 for further details.

| | Benchmarks | | | | | |
| | Multisensory model ($\sigma=3$) | | | Neuronal model | | |
| Algorithm | $\Delta$LML | MMTV | GsKL | $\Delta$LML | MMTV | GsKL |
|---|---|---|---|---|---|---|
| BBVI, MoG ($K=50$) | 1.7 [1.5,4.9] | 0.11 [0.097,0.13] | 0.17 [0.16,0.2] | 44 [33,120] | 0.6 [0.56,0.64] | 20 [17,23] |
| BBVI, MoG ($K=500$) | 1.8 [1.6,2.5] | 0.31 [0.28,0.33] | 0.53 [0.48,0.55] | 170 [140,260] | 0.67 [0.64,0.7] | 21 [18,26] |
| VBMC | **0.32** [0.23,0.37] | 0.18 [0.17,0.19] | 0.21 [0.17,0.23] | 3 [3,3.1] | 0.32 [0.31,0.32] | 140 [97,190] |
| NS (10 runs) | 0.46 [0.41,0.52] | 0.12 [0.11,0.12] | 0.072 [0.056,0.078] | 1.8 [1.8,1.9] | 0.17 [0.16,0.18] | **1.2** [0.27,1.4] |
| NS (20 runs) | 0.55 [0.5,0.59] | 0.1 [0.096,0.11] | 0.06 [0.052,0.068] | 1.8 [1.7,1.9] | 0.17 [0.15,0.18] | **1** [0.35,1.6] |
| S-VBMC (w.r.t. $\tilde{\boldsymbol{\omega}}$, 10 runs) | 0.73 [0.63,0.86] | 0.11 [0.097,0.12] | 0.062 [0.052,0.074] | 1.8 [1.7,1.8] | 0.14 [0.12,0.15] | **0.67** [0.36,1.1] |
| S-VBMC (w.r.t. $\tilde{\boldsymbol{\omega}}$, 20 runs) | 0.88 [0.86,0.95] | 0.1 [0.095,0.11] | **0.047** [0.044,0.057] | **1.5** [1.5,1.6] | **0.11** [0.087,0.13] | **0.3** [0.037,0.57] |
| S-VBMC (w.r.t. $\tilde{\mathbf{w}}$, 10 runs) | 0.93 [0.89,1] | **0.091** [0.086,0.094] | **0.042** [0.038,0.05] | 1.7 [1.6,1.7] | 0.14 [0.11,0.15] | **0.47** [0.17,0.79] |
| S-VBMC (w.r.t. $\tilde{\mathbf{w}}$, 20 runs) | 1.2 [1.2,1.3] | **0.079** [0.076,0.091] | **0.039** [0.03,0.044] | **1.5** [1.5,1.6] | **0.12** [0.092,0.13] | **0.48** [0.059,0.54] |

### A.5 Example posterior visualisations

We use *corner plots* (Foreman-Mackey, 2016) to visualise exemplar posterior approximations from different algorithms, including S-VBMC, VBMC and BBVI. These plots depict one-dimensional marginal distributions and all pairwise two-dimensional marginals of the posterior samples. Example results (chosen at random among the runs reported in Section 4 and Appendix A.4) are shown in Figures A.3, A.4, A.5, and A.6. S-VBMC consistently improves the posterior approximations over standard VBMC and generally outperforms BBVI, showing a closer alignment with the target posterior.

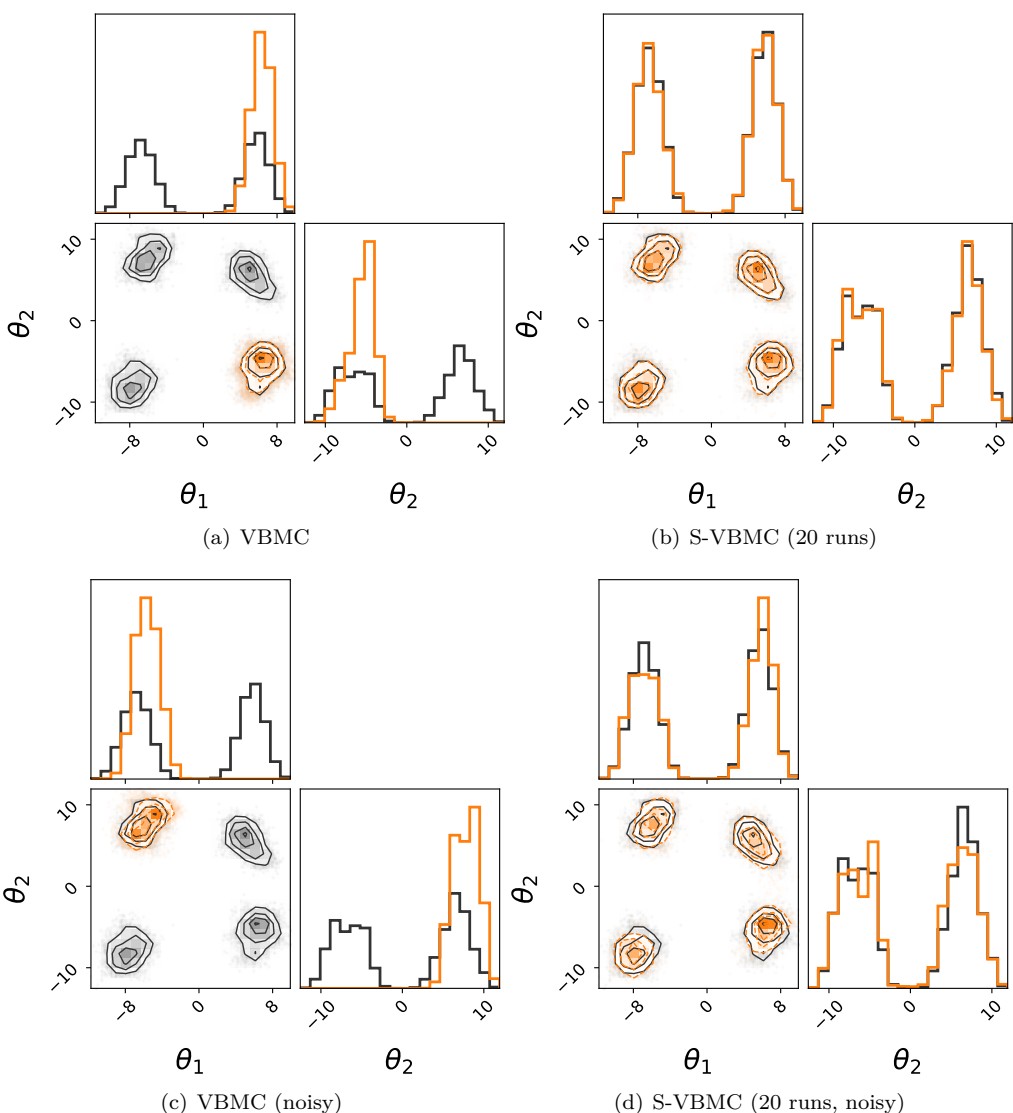

(a) VBMC

(b) S-VBMC (20 runs)

(c) VBMC (noisy)

(d) S-VBMC (20 runs, noisy)

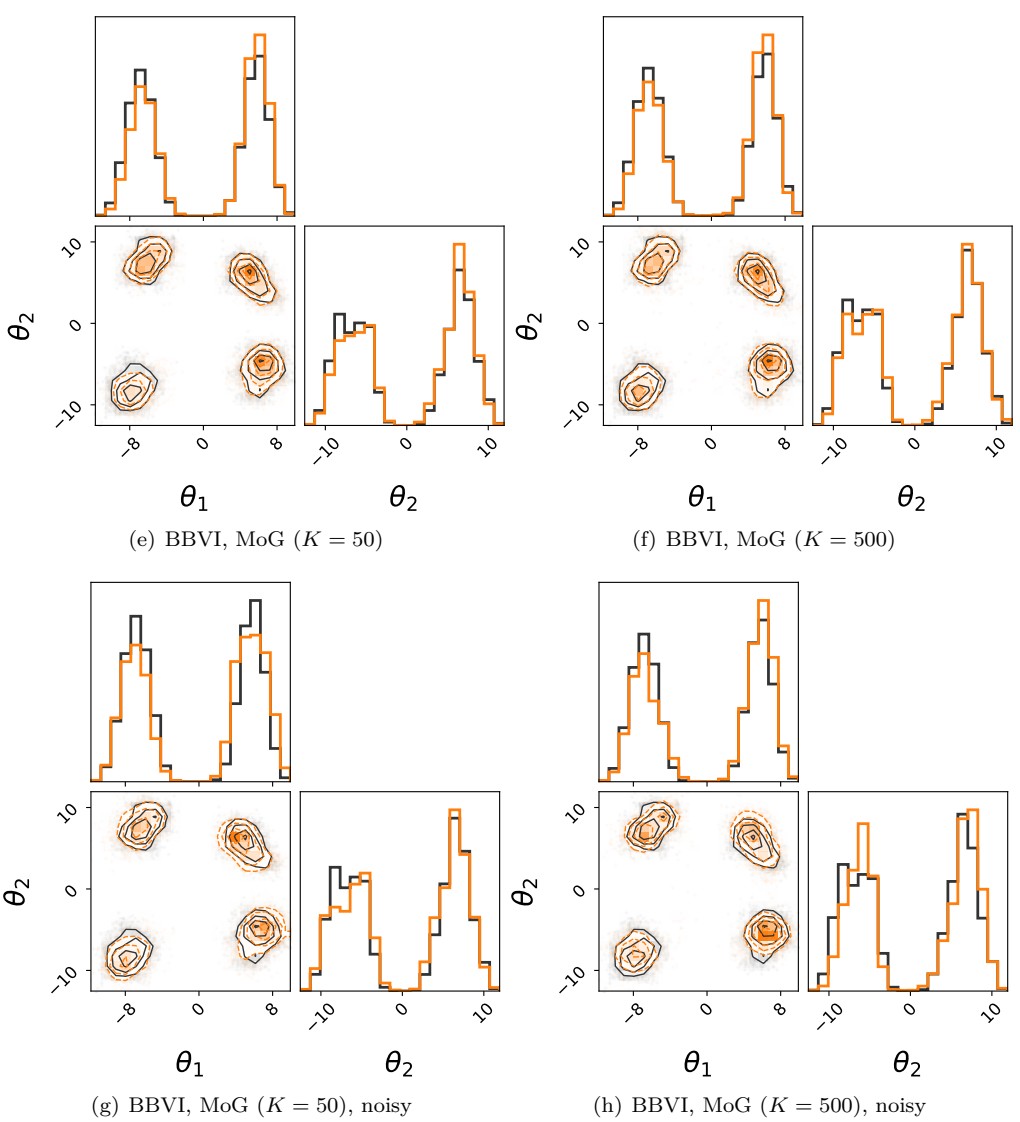

Figure A.3: GMM ($D = 2$) example posterior visualisation. Orange contours and points represent posterior samples obtained from different algorithms, while the black contours and points represent ground truth samples.

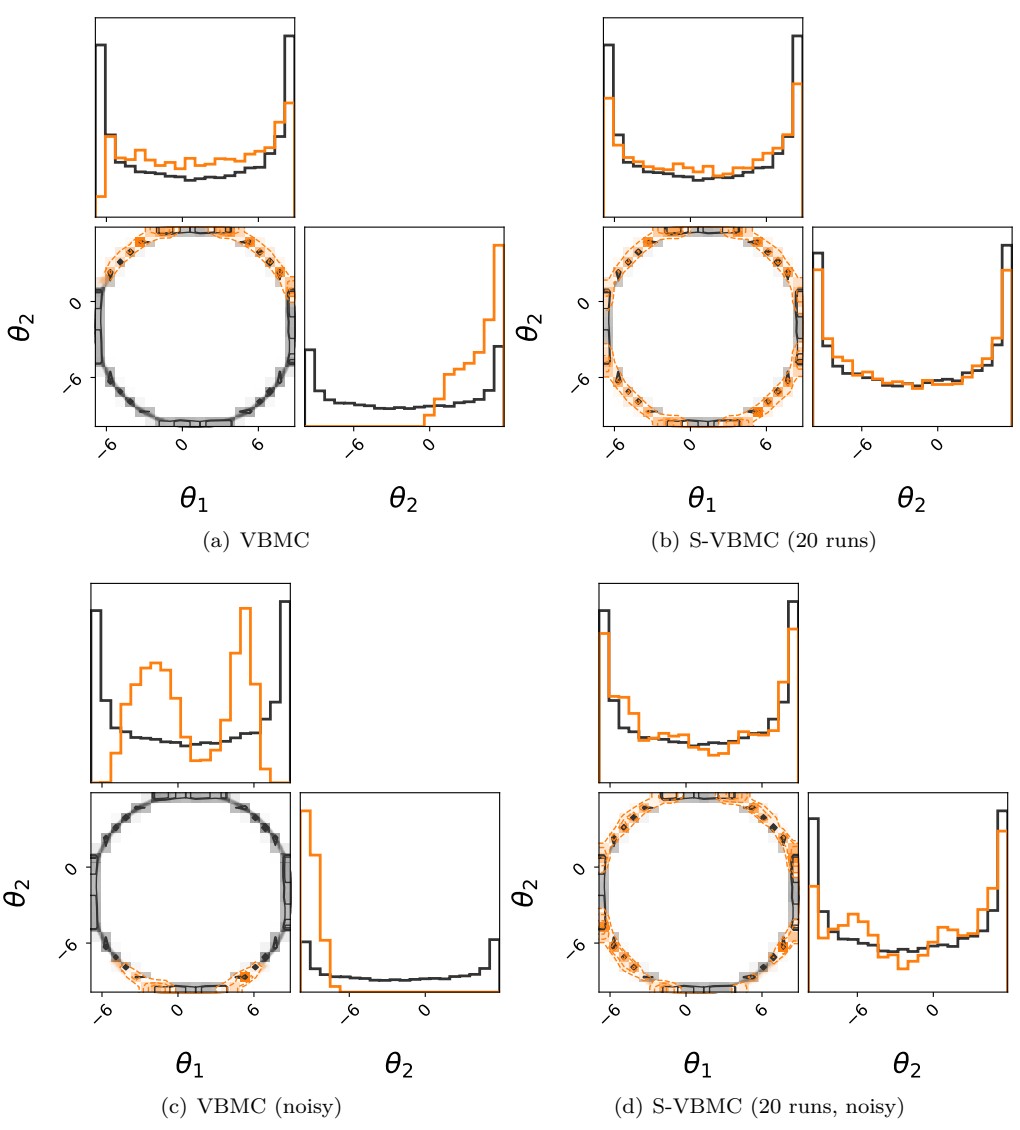

(a) VBMC

(b) S-VBMC (20 runs)

(c) VBMC (noisy)

(d) S-VBMC (20 runs, noisy)

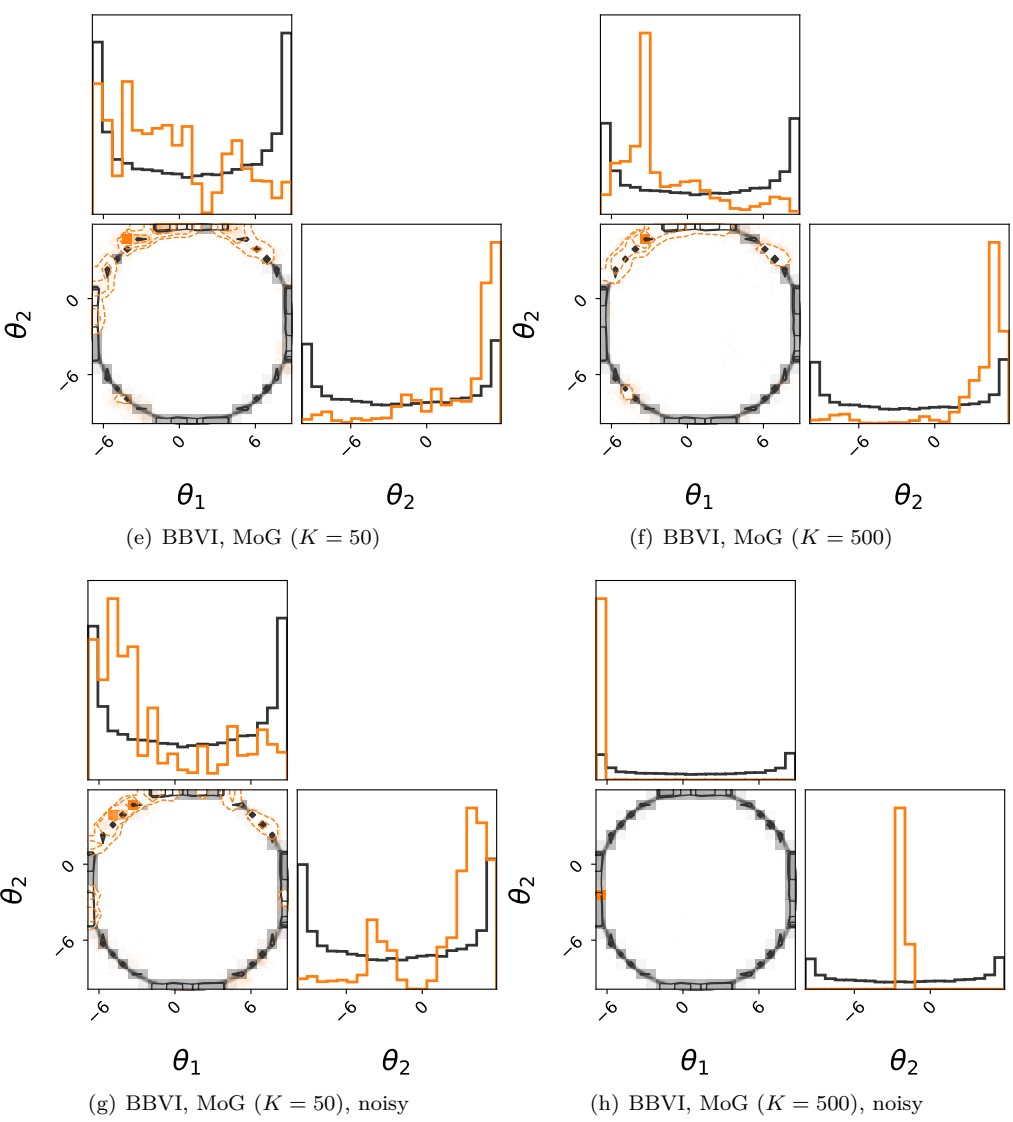

(e) BBVI, MoG ($K = 50$)      (f) BBVI, MoG ($K = 500$)

(g) BBVI, MoG ($K = 50$), noisy    (h) BBVI, MoG ($K = 500$), noisy

Figure A.4: Ring ($D = 2$) example posterior visualisation. See the caption of Figure A.3 for further details.

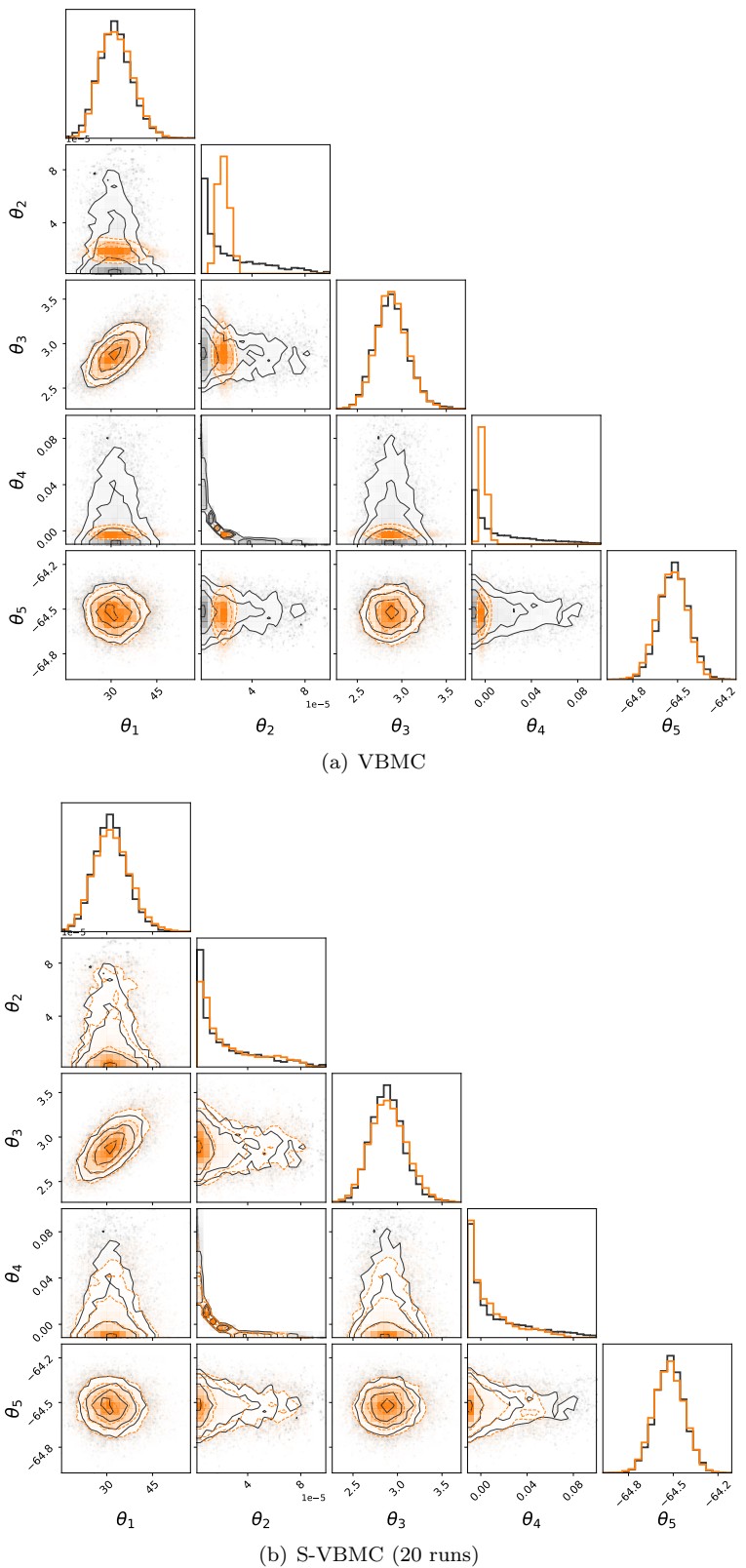

(a) VBMC

(b) S-VBMC (20 runs)

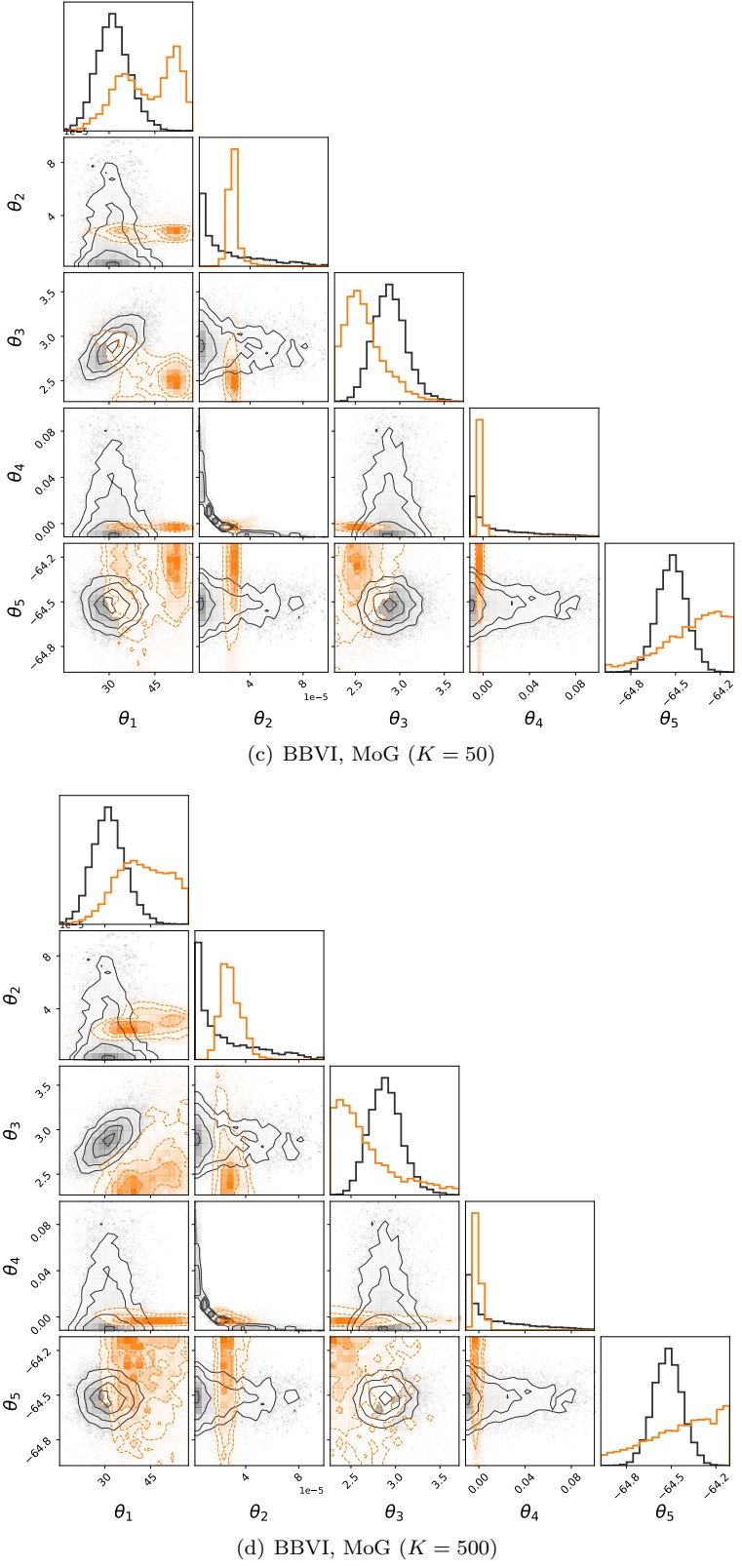

(c) BBVI, MoG ($K = 50$)

(d) BBVI, MoG ($K = 500$)

Figure A.5: Neuronal model ($D = 5$) example posterior visualisation. See the caption of Figure A.3 for further details.

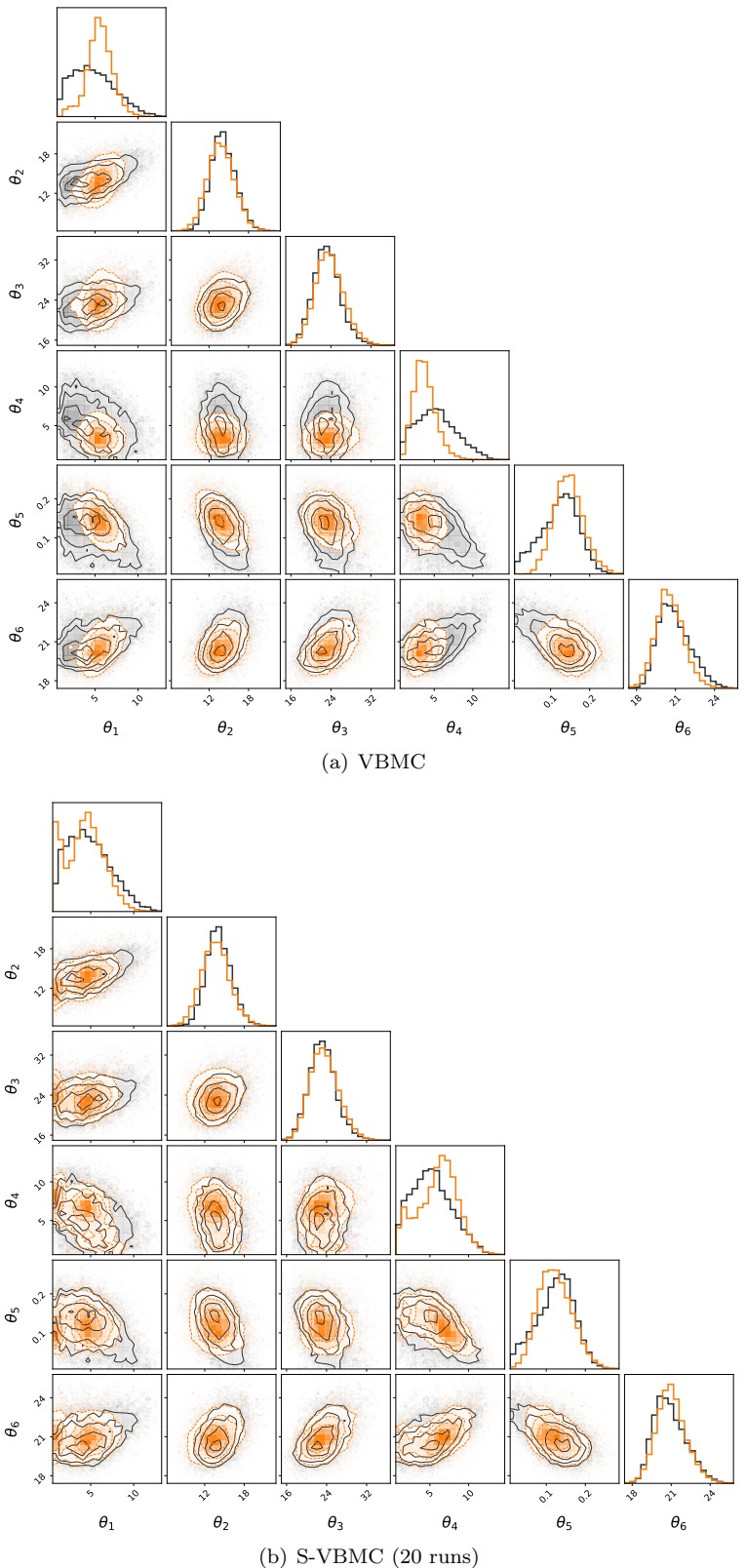

(a) VBMC

(b) S-VBMC (20 runs)

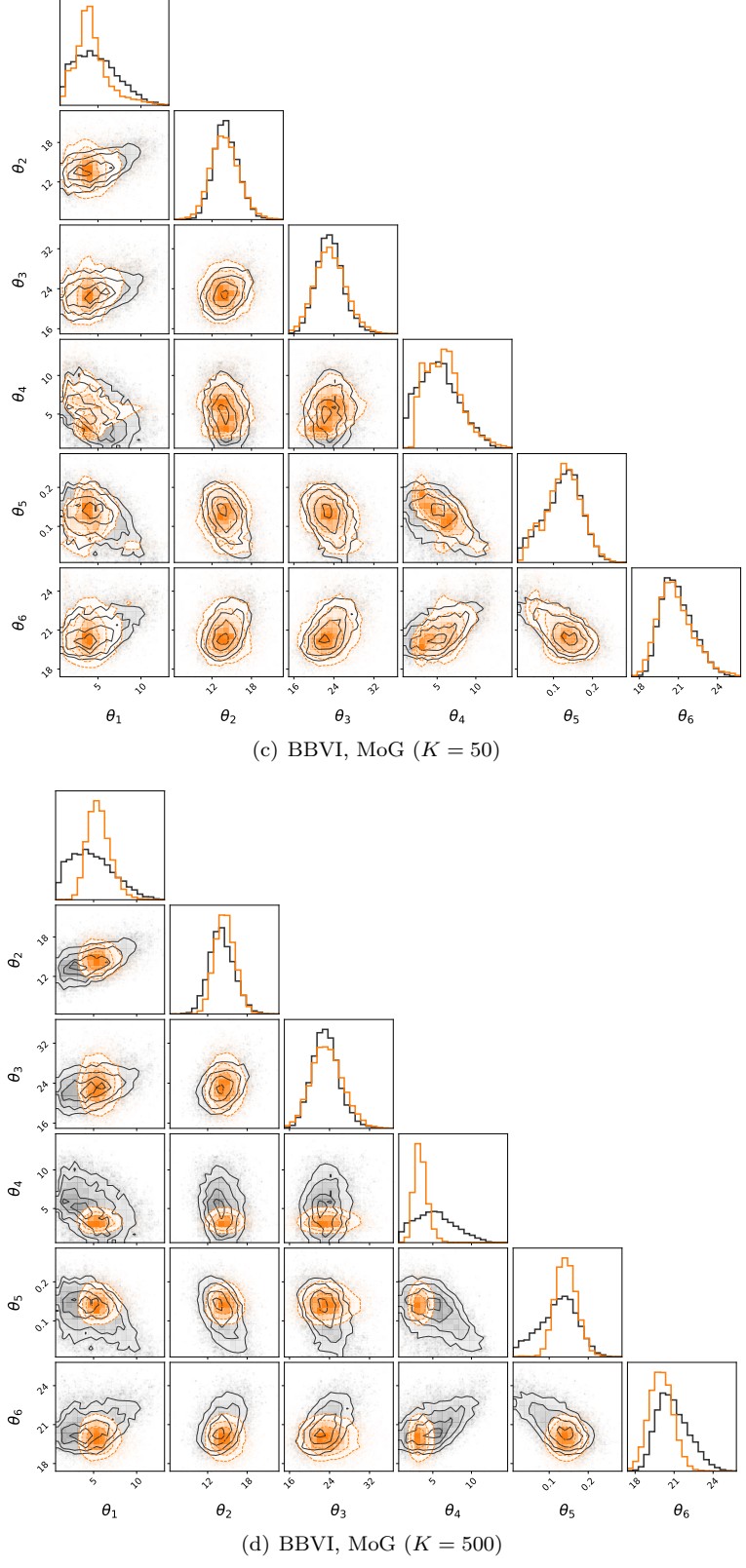

Figure A.6: Multisensory model ($D = 6, \sigma = 3$) example posterior visualisation. See the caption of Figure A.3 for further details.

