# OpenReview forum: "Stacking Variational Bayesian Monte Carlo"
_TMLR — Accepted by TMLR_

### Review · Reviewer_7953 · 2025-09-12

**Summary Of Contributions:**

This paper is interested in the setting of approximate Bayesian inference under black-box likelihoods, where it is difficult or expensive to obtain gradients of the log posterior. The paper consider the idea of variational Bayesian Monte Carlo, which trains a surrogate for the log target via variational Bayes --  more specifically, they use the idea of "stacking" to construct a global surrogate model. Stacking merges independent VBMC inference runs into a global posterior approximation and uses a mixture approximation family. The posteriors are then reweighted to obtain the final posterior estimate; this is done by optimizing a global ELBO objective w.r.t. the weights.

Strengths:
* The posterior approximation is flexible, amenable to multimodality and long tails. The approach seems very sensible as a way to solve problems with the VBMC method.
* The posterior results in comparison to the baselines seem to be more accurate.
* The manuscript is fairly clear and pleasant to read.

Weaknesses:
* I don’t see a cost vs fidelity comparison of the S-VBMC method vs the other baselines — I think these might not be directly comparable, but it would still be useful to get a sense of the relative cost, in addition to Figure 5.
* The experiments focus on lower-dimensional settings, e.g., D=1,2,5,6. While I think there are many interesting problems in this regime, I think other methods can also work well in this regime. It would be interesting to see some higher dimensional examples.
* The fidelity of the stacked posterior depends on how well the posterior can be approximated locally in many different areas; it is perhaps unclear how many local procedures need to be run in order to get a good result.

**Audience:**

Yes

**Audience Explanation:**

This paper is of interest to many communities:
* approximate Bayesian inference
* simulation-based inference
* variational inference

Thus, I think it is very relevant to a large subset of the TMLR community.

**Claims And Evidence:**

Yes

**Claims Explanation:**

I think the paper satisfies the TMLR criteria in terms correctness of claims. I think the method takes an interesting approach (VBMC) and gives a solution to how to solve a problem with its posterior approximations by using stacking.

**Requested Changes:**

Minor changes:
* I think it could be worth considering moving Figure 1 earlier -- I don't think it requires much background to understand, and it gets the point about the paper across very fast.
* One clarifying point: it seems like the original VBMC
* I would like to see more of a discussion of total cost vs other baselines besides a single VBMC run
* I'm not sure I fully understand the problem Section 5 is solving. Can this bias be expanded on in more depth, e.g., referencing specific figures and discussing the issues in more depth?
* In practice, how should one choose the number of components and the number of posteriors?
* It seems like the method is meant to focus on lower-dimensional posteriors, due to the dependence of VBMC on Gaussian process surrogates and Bayesian quadrature; perhaps this can be clarified and/or made more explicit earlier in the paper, e.g., in the introduction, especially for readers who might be unfamiliar with VBMC.

---

> ### Author Response · Authors · 2025-09-29
> **Rebuttal by Authors (1)**
>
> Thank you for your appreciative review of our paper and your thoughtful comments. In the spirit of TMLR’s brief, to-the-point review process, we directly respond below to the requested changes.
>
> > I think it could be worth considering moving Figure 1 earlier -- I don't think it requires much background to understand, and it gets the point about the paper across very fast.
>
> Great suggestion – indeed it is a good explanatory figure that can intuitively introduce the reader to the method much earlier in the paper. We will move the figure to Section 1 (Introduction), slightly updating the text and caption as needed.
>
>
> > One clarifying point: it seems like the original VBMC
>
> The text here appears to be incomplete. We are happy to clarify should the doubt still stand after our response to reviewers.
>
>
> > I would like to see more of a discussion of total cost vs other baselines besides a single VBMC run
>
> Please note that, as is common in the “surrogate-based” literature, from Bayesian Optimization (Snoek et al., 2012; Garnett, 2023) to surrogate-based inference (Acerbi, 2018; Wang & Li, 2018; Acerbi, 2020; Järvenpää et al., 2021; El Gammal et al., 2023; Järvenpää & Corander, 2024), we demonstrated our method on several problems where function evaluations are not computationally expensive, for the simple reason that a full benchmark with multiple actually expensive models is impractical. So wall-clock time needs to be interpreted carefully as a metric when comparing methods with different likelihood evaluation costs.
>
> We can directly compare VBMC and S-VBMC because by construction they use the same backbone method and have the same evaluation costs (S-VBMC adds a small postprocessing cost, which, crucially, does not depend on the cost of likelihood evaluation). Conversely, comparing to non-VBMC methods becomes highly problem-dependent. The typical solution would consist of matching the **number of function evaluations**, for which non-surrogate-based baselines would be at a significant disadvantage, as demonstrated in previous work (Acerbi, 2018; 2020).
>
> Having said that, with the important caveat above, we are happy to add the following table to Appendix 2 (Additional Experiments), which reports the **total runtime cost** of BBVI for various problems. As expected, BBVI is much faster than (S-)VBMC on problems with fast likelihood evaluation, but as soon as the likelihood becomes even slightly expensive – $\approx 0.7$ seconds per evaluation for the neuronal model – the cost on non-VBMC methods skyrockets – demonstrating the kind of scenarios VBMC was developed to solve in the first place.
>
> *Table A.1. BBVI runtime (in seconds) compared to that of 40 (parallel) VBMC runs and their subsequent stacking with S-VBMC. Values show median with 95% confidence intervals (computed from 10000 bootstrap resamples) in brackets. Bold entries indicate the best median performance (i.e., lowest compute time).*
>
> | Task                              | BBVI runtime (s)                       | VBMC + S-VBMC (40 runs) runtime (s)        |
> |-----------------------------------|------------------------------|---------------------------------|
> | GMM (noiseless)                   | **9.9** [9.4, 10.3]          | 458.3 [428.3, 510.8]            |
> | GMM (σ = 3)                       | **12.8** [12.2, 13.7]        | 857.8 [759.0, 954.8]            |
> | Ring (noiseless)                  | **12.2** [11.7, 12.6]        | 559.3 [516.9, 794.2]            |
> | Ring (σ = 3)                      | **14.0** [13.4, 15.0]        | 1269.9 [1206.0, 1557.4]         |
> | Neuronal model (noiseless)        | 8497.0 [8411.2, 8617.8]      | **1561.8** [1445.1, 1900.1]     |
> | Multisensory model (σ = 3)        | **24.5** [21.2, 27.3]        | 3149.5 [2616.3, 3525.1]         |

---

> ### Author Response · Authors · 2025-09-29
> **Rebuttal by Authors (2)**
>
> > I'm not sure I fully understand the problem Section 5 is solving. Can this bias be expanded on in more depth, e.g., referencing specific figures and discussing the issues in more depth?
>
> The problem discussed in Section 5 is an overestimation of the ELBO (defined in Eq. 2) that results from stacking VBMC posteriors. This is visible in Figure 3 (b) and (d), first column, and Figure 4 (b), first column, in that the estimated ELBO from S-VBMC on these problems slightly “overshoots” the ground truth (the S-VBMC estimates, blue dots, end **above** the black horizontal line). What characterises these problems, and is a main source of this bias, is the fact that all these problems use a stochastic estimator for the likelihood (as opposed to exact likelihood evaluations used in the other problems, where the estimate does not overshoot).
>
> To expand on this, as mentioned in Section 2.2, the ELBO is **always** lower than the log marginal likelihood (LML), with equality when the approximate posterior perfectly corresponds to the true posterior. However, as shown in Figures 3 and 4, in our results the estimated ELBO for problems with noisy likelihoods grows larger than the ground-truth LML. As this cannot be true, there must be a positive bias in the ELBO estimate. As one can see on the figures (first column), this bias builds up as more and more VBMC runs are stacked.  This does not affect the quality of the posterior (which keeps improving or plateaus), but it is a potential issue if one wants to use the estimated ELBO for Bayesian model comparison.
>
> In Section 5.1 we propose a candidate origin for this bias, and in Section 5.2 we introduce a simple heuristic to counteract it, which worked well in our experiments.
> We will expand the very first paragraph of Section 5 (i.e. before Section 5.1) to re-iterate how the ELBO cannot be larger than the LML and refer more clearly to the results in Section 4 (and Figures 3 and 4) to give the reader more context about what the issue is.
>
>
> > In practice, how should one choose the number of components and the number of posteriors?
>
> The main hyperparameter in S-VBMC is the **number of posteriors** $M$ (distinct VBMC runs). In Section 4.4 and 4.5, we point out that most of the gains in our metrics happen when we stacked up to 10 VBMC runs ($M = 10$), with relatively small additional improvements when stacking more. Furthermore, in Section 4.6, we discuss how stacking 10 runs vastly improves the posterior with relatively small ($\approx 5-15 \%$) computational overhead. Therefore, our experiments showed $M=10$ to be a nice “sweet spot” where the posterior improves greatly and the overhead remains contained. That said, there is no harm in stacking more runs (in our experiments, that never hurt posterior quality), but a user should be mindful of the increasing compute time.
> We will add a reminder of this in our Discussion (Section 6), where we discuss our overhead analysis.
>
> As for the **maximum number of components**, this is set by the VBMC algorithm (Acerbi 2018; 2020), and we used the default – and recommended – settings of the `pyvbmc` package (Huggins et al., 2023) for all our experiments (mentioned in Section 4.1). This results in a final number of 50 components per VBMC posterior, so that the total number of components for all our experiments was $M \times 50$.
>
>
> > It seems like the method is meant to focus on lower-dimensional posteriors, due to the dependence of VBMC on Gaussian process surrogates and Bayesian quadrature; perhaps this can be clarified and/or made more explicit earlier in the paper, e.g., in the introduction, especially for readers who might be unfamiliar with VBMC.
>
> Thank you for the suggestion. Yes, this is a well-known and clear limitation of all these methods, widely known to VBMC users (Huggins et al., 2023). We agree that the reference to the low-dimensional setting $D \le \approx 10$ should be reiterated clearly in the introduction. We will add this.

---

> > ### Author Response · Authors · 2025-09-30
> > **Rebuttal by Authors (3)**
> >
> > **References**
> >
> > Acerbi, L. (2018). Variational Bayesian Monte Carlo. Advances in Neural Information Processing Systems, 31, 8222-8232.
> >
> > Acerbi, L. (2020). Variational Bayesian Monte Carlo with noisy likelihoods. Advances in neural information processing systems, 33, 8211-8222.
> >
> > El Gammal, J., Schöneberg, N., Torrado, J., & Fidler, C. (2023). Fast and robust Bayesian inference using Gaussian processes with GPry. Journal of Cosmology and Astroparticle Physics, 2023(10), 021.
> >
> > Garnett, R. (2023). Bayesian optimization. Cambridge University Press.
> >
> > Huggins, B., Li, C., Tobaben, M., Aarnos, M. J., & Acerbi, L. (2023). PyVBMC: Efficient Bayesian inference in Python. Journal of Open Source Software 8(86), 5428.
> >
> > Järvenpää, M., & Corander, J. (2024). Approximate Bayesian inference from noisy likelihoods with Gaussian process emulated MCMC. Journal of Machine Learning Research, 25(366), 1-55.
> >
> > Järvenpää, M., Gutmann, M. U., Vehtari, A., & Marttinen, P. (2021). Parallel Gaussian process surrogate Bayesian inference with noisy likelihood evaluations. Bayesian Analysis, 16(1), 147–178.
> >
> > Snoek, J., Larochelle, H., & Adams, R. P. (2012). Practical Bayesian optimization of machine learning algorithms. Advances in neural information processing systems, 25, 2951–2959.
> >
> > Wang, H., & Li, J. (2018). Adaptive Gaussian process approximation for Bayesian inference with expensive likelihood functions. Neural computation, 30(11), 3072-3094.

---

### Review · Reviewer_yRB2 · 2025-09-14

**Summary Of Contributions:**

In many applied probabilistic models, the likelihood is difficult or expensive to evaluate, which makes the Bayesian posterior hard to obtain using the off-the-shelf methods. Variational Bayesian Monte Carlo (VBMC) provides an elegant solution: the unnormalized posterior is approximated by a surrogate Gaussian process (GP) in a similar way to Bayesian optimization, and then variational inference with mixture distributions is performed to approximate the surrogate model. This approximation is efficient with analytical computations with the GP. With VBMC, it is also possible to control the number of model evaluations to construct the GP model according to the budget.

However, the approximation in VBMC is not perfect. In practice it may not cover the tails of long-tailed distributions, and it can miss areas in distributions with multiple modes. Bayesian stacking was invented to address these issues. This work proposes S-VBMC, which borrows ideas from Bayesian stacking to construct the approximate posterior with a mixture of imperfect posteriors with multiple VBMC runs. The pipeline has two phases: (1) M parallel VBMC instances are performed; (2) the mixing weights of the M posteriors are obtained using variational inference.

In the experiments, it is demonstrated that S-VBMC outperforms BBVI, VBMC and naive stacking on both synthesis and real data measured by multiple metrics. In addition, if phase (1) is performed in perfect parallel (which is achievable), the computing overhead from phase (2) is small compared to phase (1). The experiments also reveal a bias in the pipeline: as M is increased, the error to the true posterior does not goes to zero.

This work then explores reasons to explain the bias and ways to mitigate it. The intuition is that VBMC produces approximate models randomly, and stacking tends to select mixture components that overestimate ELBO. Then, capping techniques are introduced to correct the bias, which are found useful in the experiments.

**Audience:**

Yes

**Audience Explanation:**

There are different assumptions in Bayesian computation. The proposed method is suitable for models with expensive likelihood evaluations. Applied researchers may find the method interesting when combined with their works.

Another interesting aspect is the parallel computation of Bayesian pipeline. Though not explicitly stated in the manuscript, this method has the potential of using large scale computing power. It is possible to develop better softwares that utilize modern hardwares with the method.

The negative result - the bias of stacking of random objectives, is particularly educational. Now people have been using approximate algorithms for intractable problems. The issues this work met may happen to other ideas. I think there may be interesting statistical problems in this space.

Nevertheless, the scope of the works is narrow. The work claims "could theoretically extend to other variational approaches based on mixture posteriors", but even if the claim is correct, a method that helps variational method with mixture posteriors is not very generalizable. Unless there are more discussions or experiments in this direction, I think this is the major limitation.

**Broader Impact Concerns:**

I don't perceive potential broader impact concerns.

**Claims And Evidence:**

Yes

**Claims Explanation:**

(1) With synthesis data, this work shows the problems of VBMC and how S-VBMC addresses the problems. With real data, there are also enough empirical evidences to compare the proposed methods with existing approaches. The work also experiments with the computation overhead, which is valuable to know about.

(2) To demonstrate the problems of overestimation, the work uses a simplified example but the example is representative of common scenarios. I think the example is a clear theoretical evidence and motivates the solutions well.

**Requested Changes:**

Throughout the work there is a variable $K_m$ that controls the number of mixture components per VBMC run. But the selection of this variable, and how it affects the results, are not discussed. Is there a reason to vary $K_m$ for different $m$? How are they selected in the experiments? I'd like to understand the questions better. Specifally in (12), it also assigns higher initial weights to posteriors with smaller $K_m$, which encourages worse settings. Probably it is a design choice, but there should be some discussion about why $K_m$ is in the denomenator.

I'd like to see the hardware specification of the experiments. It would be particular interesting if there are GPU experiments that demonstrate the scalability of the methods.

---

> ### Author Response · Authors · 2025-09-29
> **Rebuttal by Authors**
>
> Thank you for your insightful comments and your overall positive review. In what follows we address one by one the points you raised in the “Requested changes” section of your review.
>
> > Throughout the work there is a variable $K_m$ that controls the number of mixture components per VBMC run. But the selection of this variable, and how it affects the results, are not discussed. Is there a reason to vary $K_m$ for different $m$? How are they selected in the experiments? I'd like to understand the questions better. Specifally in (12), it also assigns higher initial weights to posteriors with smaller $K_m$, which encourages worse settings. Probably it is a design choice, but there should be some discussion about why $K_m$ is in the denomenator.
>
> In our experiments, when running VBMC, we used its default settings, which assign a maximum of $K = 50$ components per approximate posterior. We did not experiment with VBMC settings and used it “out of the box”, as these settings were chosen and experimentally validated by the VBMC developers in a series of papers (Acerbi 2018, 2019, 2020). Generally speaking, the VBMC developers recommend not to change the default VBMC settings as the inner workings of individual VBMC runs are optimised for the default choice (Huggins et al., 2023). Thus, our paper focuses on how to merge the results of $M$ independent VBMC runs, and $M$ is the key hyperparameter of our method, which we test extensively in the paper. We will slightly edit Section 4.1 (Procedure) to point out that all the VBMC posteriors in our experiments had 50 Gaussian components.
>
> Finally, thank you for pointing out the denominator in Eq. 12 (weight initialisation). That is indeed not needed, as weighting by $w_{m, k}$ already ensures that each individual VBMC posterior receives an initial weight proportional to its corresponding $\exp(\text{ELBO}(\boldsymbol{\phi}_m))$. We will remove it from the revised manuscript. In practice, as in our experiments $K_m = 50$ in all cases (see above), this did not have any impact on our results.
>
>
>
>
> > I'd like to see the hardware specification of the experiments. It would be particular interesting if there are GPU experiments that demonstrate the scalability of the methods.
>
> Thank you for raising this point about scalability. Our method is designed to be computationally lightweight by construction, and scalability is not a major concern.
>
> Regarding hardware specifications, we ran all our experiments on CPU, specifically AMD EPYC 7452 Processor, with 16GB of RAM. We will add these hardware specifications to Section 4.1 (Procedure) in the revised manuscript.
>
> S-VBMC can use GPU acceleration (being written in standard PyTorch), but it’s a very light post-processing step, with only a few hundred parameters (or up to 2000 in our experiments) being optimised. Our results show that stacking 10 runs (and thus optimising 500 parameters) is sufficient in most cases, so there would be no practical need to stack a number of posteriors for which GPU acceleration would actually make a difference.
>
> As for VBMC, the runs required for our approach are independent and thus can be run in an “embarrassingly parallel” setup, so that scalability would not be an issue here either. Furthermore, VBMC is currently written in Numpy/Scipy (Huggins et al., 2023), so it does not use GPU acceleration. The main bottleneck of VBMC is assumed to be the expensive models being evaluated. Future work might expand VBMC to modern packages with GPU support (e.g., GPyTorch, Gardner et al., 2018; BoTorch, Balandat et al., 2020), but this is a task for the VBMC developers and beyond the scope of our current submission.
>
>
>
> **References**
>
> Acerbi, L. (2018). Variational Bayesian Monte Carlo. Advances in Neural Information Processing Systems, 31, 8222-8232.
>
> Acerbi, L. (2019). An exploration of acquisition and mean functions in Variational Bayesian Monte Carlo. Proceedings of The 1st Symposium on Advances in Approximate Bayesian Inference (PMLR) 96, 1–10.
>
> Acerbi, L. (2020). Variational Bayesian Monte Carlo with noisy likelihoods. Advances in neural information processing systems, 33, 8211-8222.
>
> Balandat, M., Karrer, B., Jiang, D., Daulton, S., Letham, B., Wilson, A. G., & Bakshy, E. (2020). BoTorch: A framework for efficient Monte-Carlo Bayesian optimization. Advances in neural information processing systems, 33, 21524-21538.
>
> Gardner, J., Pleiss, G., Weinberger, K. Q., Bindel, D., & Wilson, A. G. (2018). GPyTorch: Blackbox matrix-matrix Gaussian process inference with GPU acceleration. Advances in neural information processing systems, 31, 7587–7597.
>
> Huggins, B., Li, C., Tobaben, M., Aarnos, M. J., & Acerbi, L. (2023). PyVBMC: Efficient Bayesian inference in Python. Journal of Open Source Software 8(86), 5428.

---

### Review · Reviewer_vbeX · 2025-09-16

**Summary Of Contributions:**

The paper considers a variational inference problem, where the likelihood evaluation is assumed to be expensive.
Authors considers the variational Bayesian Monte Carlo (VBMC) algorithm, wherein an posterior approximation by a Gaussian mixture is constructed by modelling the log joint using a Gaussian process, and estimating the ELBO with the Bayesian quadrature.

The main contribution of the paper is investigating an algorithm, called S(tacked)-VBMC, that combines a mixture of Gaussian mixtures from independent runs of VBMCs, in order to produce a better approximation to the posterior.

Strengths:
* The algorithm basically builds on top of any estimator of the log joint associated with a given variational family. The use of VBMC is supported by an existing software, and relevant to the blackbox setting.
* The algorithm is simple to implement since it just requires an optimisation of the mixing weights that combine the learned Gaussian mixtures from the independent runs of VBMCs.

Weaknesses:
* As there is no communication between VBMC runs, it is unclear if one can obtain an improved estimate, since VBMC runs might end up exploring the same region, and produce similar variational approximations. Commenting on how to set up the runs would be useful.

**Additional Comments:**

The proposed algorithm does not take advantage of the variance of the integral estimate (I think this is the point using a Bayesian quadrature). Taking into account the uncertainty could perhaps be useful for countering the overestimation issue; for example, we could consider not giving too much of weights to those with high uncertainty.

**Audience:**

Yes

**Audience Explanation:**

The expensive likelihood setting is one of the challenging areas for applications (e.g., models with expensive simulators), and thus the work is aligned with TMLR's audience scope.

**Claims And Evidence:**

Yes

**Claims Explanation:**

Overall, the paper shows the merit of performing the post processing of VBMC runs with through experiments.

The proposed algorithm is compared with BBVI by Ranganath et al, 2014, a standard benchmark, and a naive stacking where all Gaussian mixtures are given equal weights.
It is shown that while it is simple to implement, the proposed method outperforms these baselines in performance metrics such as ELBO.

The authors also acknowledge an issue with the proposed method: it tends to overestimate the ELBO (and the marginal likelihood) as the number of VMBC runs increases, when the evaluation of the log joint involves errors.
The authors propose a heuristic approach (capping the log joint estimate using an statistic of integral estimates) to counter this, and shows that it somewhat remedies this issue.

**Requested Changes:**

* The notation for the integral of the log joint and its estimator should be distinct, as it is confusing. For example, the hat notation is used in Eq. 24. Likewise, the RHS of Eq.8 should be given a symbol like $\hat{I}_k$, since this is the $\mathbf{I}_m$ vector that appears in Section 3 (the integral o the LHS is unavaiable). Other parts such as Eq. 26 should also be fixed,
* Adam does not take into account the constraint on the weights (i.e., they form a probability vector). I would like a comment on this--how do the authors ensure this constraint? An easy fix would be to reparametrise the weights (see, e.g., [this paper](https://www.jmlr.org/papers/v26/23-1166.html)).
* Having a sentence like "we consider the following two candidates for $E_{\mathrm{cap}}$ under Eq. 28 would help the transition.
* It is unclear the computational cost of  a single run of VBMC is matched with S-VBMC -- can one achieve the same performance given the same computational cost with sufficiently flexible Gaussian mixtures? If this is challenging, it would highlight the point of using the proposed method.

---

> ### Author Response · Authors · 2025-09-29
> **Rebuttal by Authors**
>
> Thank you for your review and for appreciating our contribution, as well as for your suggestions specified in the “Requested changes” section. Here we address them one by one.
>
> > The notation for the integral of the log joint and its estimator should be distinct, as it is confusing. For example, the hat notation is used in Eq. 24. Likewise, the RHS of Eq.8 should be given a symbol like $\hat{I}_K$, since this is the $\mathbf{I}_m$ vector that appears in Section 3 (the integral o the LHS is unavaiable). Other parts such as Eq. 26 should also be fixed,
>
> Thank you for pointing this out. We agree that the notation is confusing in this case. We will correct it so that $\hat{I}_k$ clearly refers to the VBMC estimate, and $I_k$ to the true (unknown) value.
>
>
> > Adam does not take into account the constraint on the weights (i.e., they form a probability vector). I would like a comment on this--how do the authors ensure this constraint? An easy fix would be to reparametrise the weights (see, e.g., this paper).
>
> Thanks – we used standard solutions in the probabilistic inference community. Specifically, we reparametrised the weights to optimise unconstrained logits and applied the softmax function when evaluating the objective ($\text{ELBO}_{\text{stacked}}$) to map them to simplex weights. This adds one undetermined and irrelevant degree of freedom which does not affect the optimisation (regularisation can be added in extreme cases to avoid numerical instability, but it’s not needed in our case). Thank you for pointing out this information is missing (as did Reviewer ajye), we will specify this in Section 3.
>
>
> > Having a sentence like "we consider the following two candidates for $E_{cap}$ under Eq. 28 would help the transition.
>
> Thank you for the suggestion, we will add that in.
>
>
> > It is unclear the computational cost of a single run of VBMC is matched with S-VBMC -- can one achieve the same performance given the same computational cost with sufficiently flexible Gaussian mixtures? If this is challenging, it would highlight the point of using the proposed method.
>
>
> Thank you for this important point, also raised by Reviewer ajye. We investigated whether a single VBMC run with more components could match S-VBMC's performance.
>
> In practice, we found that VBMC fails to converge when $K_m$ exceeds $\approx 500$ components (equivalent to our S-VBMC with $M=10$), with greatly increased runtime even approaching this limit. The reason is that the VBMC algorithm uses a dynamic procedure to set the number of components at runtime, based on the maximum number of components (see Acerbi, 2018). VBMC was carefully optimised for its default settings ($K=50$) through extensive validation (Acerbi, 2018; 2019; 2020), and the developers explicitly recommend against departing from these settings (Huggins et al., 2023).
>
> Importantly, VBMC's failures in our benchmarks stem not from insufficient mixture components – 50 components can approximate most targets well – but from its conservative exploration strategy and limited evaluation budget, which produce good local but not global approximations for complex targets. S-VBMC's strength lies in combining multiple good local approximations (obtained in parallel) into a robust global approximation, leveraging the benefits of diverse initialisations and parallelisation that would be lost by simply increasing the evaluation budget of a single run.
>
> A thorough investigation of VBMC's behavior with non-standard settings (e.g., $K = 500$) would require substantial modifications to the algorithm's inner workings, which is beyond the scope of this work.
>
>
> **References**
>
> Acerbi, L. (2018). Variational Bayesian Monte Carlo. Advances in Neural Information Processing Systems, 31, 8222-8232.
>
> Acerbi, L. (2019). An exploration of acquisition and mean functions in Variational Bayesian Monte Carlo. Proceedings of The 1st Symposium on Advances in Approximate Bayesian Inference (PMLR) 96, 1–10.
>
> Acerbi, L. (2020). Variational Bayesian Monte Carlo with noisy likelihoods. Advances in neural information processing systems, 33, 8211-8222.
>
> Huggins, B., Li, C., Tobaben, M., Aarnos, M. J., & Acerbi, L. (2023). PyVBMC: Efficient Bayesian inference in Python. Journal of Open Source Software 8(86), 5428.

---

### Review · Reviewer_ajye · 2025-09-20

**Summary Of Contributions:**

The paper addresses approximate Bayesian inference in settings where the likelihood is expensive to evaluate, making standard posterior inference methods infeasible. It builds on the variational Bayesian Monte Carlo (VBMC) framework, which uses a Gaussian process surrogate to model the log joint and estimates the ELBO with Bayesian quadrature. The main contribution is S-VBMC, an approach that combines multiple independent VBMC runs using a stacking procedure, yielding a mixture of mixtures posterior approximation. This design aims to improve robustness in capturing multimodality and long-tailed distributions, while remaining compatible with black-box likelihoods.

Strengths

* The idea of combining multiple VBMC runs through stacking is conceptually simple and easy to implement, requiring only an optimization of mixture weights over existing VBMC outputs.
* The method is flexible and can capture multimodal and long-tailed posterior structures better than a single VBMC run.
* The approach is naturally parallelizable: the bulk of the computation lies in independent VBMC runs, with the stacking step adding minimal overhead.
* Experiments demonstrate competitive performance compared to BBVI, VBMC, and naive stacking baselines.

Weaknesses:

* The procedure to optimize the weights on the simplex is not well detailed.
* The impact of the number of VBMC runs (posteriors) and the number of mixture components within each posterior is not well explored, leaving unclear intuition on how these choices affect performance.
* Since VBMC runs are independant, there is a risk of redundancy if multiple runs converge to similar local approximations.
* The empirical evaluation is mostly limited to low-dimensional settings; it remains unclear how well the method scales to higher-dimensional problems.

**Audience:**

Yes

**Audience Explanation:**

The findings address a core challenge in approximate Bayesian inference with expensive likelihoods, a problem of broad interest to the TMLR audience. The proposed method is relevant to both methodological researchers and practitioners, as it improves posterior approximation while remaining practical and broadly applicable.

**Claims And Evidence:**

Yes

**Claims Explanation:**

The claims are supported by accurate and convincing evidence, with experiments conducted in settings relevant to approximate Bayesian inference under expensive likelihoods. The results show clear improvements over baselines, and the paper also acknowledges and analyzes limitations, such as bias from over-estimation.

**Requested Changes:**

Several points could strengthen the paper:
* While the naive stacking and "posterior-only" variants are useful ablations, it would also be valuable to compare against a baseline where VBMC is run directly with $\sum_{m=1}^M K_m$ components, to better understand trade-offs in cost and performance.
* Since the weight vector lies on the simplex, its optimization cannot rely on standard SGD. A more detailed explanation of the optimization procedure would help clarify this step.
* The impact of both the number of posteriors and the number of components per posterior is not deeply discussed, yet this information would be important for practitioners. Related to this, it would be insightful to analyze how stacking handles poor-quality posteriors and whether the weight optimization effectively downweights them.
* Finally, additional experiments in higher dimensions would be highly informative. As the authors acknowledge, dimensionality is a known limitation of VBMC. It would be interesting to see whether stacking can extend its applicability in this regard.

---

> ### Author Response · Authors · 2025-09-29
> **Rebuttal by Authors (1)**
>
> Thank you for your insightful review and for appreciating our work. Here we respond to the concerns you raised in the “Requested changes” section of your review, point by point.
>
> > While the naive stacking and "posterior-only" variants are useful ablations, it would also be valuable to compare against a baseline where VBMC is run directly with $\sum_{m=1}^M K_m$ components, to better understand trade-offs in cost and performance.
>
> Thank you for this important point, also raised by Reviewer vbeX. We investigated whether a single VBMC run with a number of components matching the total number of components in S-VBMC could match S-VBMC's performance.
>
> In practice, we found that VBMC fails to converge when $K_m$ exceeds $\approx 500$ components (equivalent to our S-VBMC with $M=10$), with greatly increased runtime even approaching this limit. The reason is that the VBMC algorithm uses a dynamical procedure to set the number of components at runtime, based on the maximum number of components (see Acerbi, 2018). VBMC was carefully optimised for its default settings ($K=50$) through extensive validation (Acerbi, 2018; 2019; 2020), and the developers explicitly recommend against departing from these settings (Huggins et al., 2023).
>
> Importantly, VBMC's failures in our benchmarks stem not from insufficient mixture components – 50 components can approximate most targets well – but from its conservative exploration strategy and limited evaluation budget, which produce good local but not global approximations for complex targets. S-VBMC's strength lies in combining multiple good local approximations (obtained in parallel) into a robust global approximation, leveraging the benefits of diverse initialisations and parallelisation that would be lost by simply increasing the evaluation budget of a single run.
>
> A thorough investigation of VBMC's behavior with non-standard settings (e.g., $K = 500$) would require substantial modifications to the algorithm's inner workings, which is beyond the scope of this work.
>
>
>
> > Since the weight vector lies on the simplex, its optimisation cannot rely on standard SGD. A more detailed explanation of the optimisation procedure would help clarify this step.
>
>
>
> Thanks – we used standard solutions in the probabilistic inference community. Specifically, we reparametrised the weights to optimise unconstrained logits and applied the softmax function when evaluating the objective ($\text{ELBO}_{\text{stacked}}$ to map them to simplex weights. This adds one undetermined and irrelevant degree of freedom which does not affect the optimisation (regularisation can be added in extreme cases to avoid numerical instability, but it’s not needed in our case). Thank you for pointing out that this information is missing (as did Reviewer vbeX), we will specify this in Section 3.

---

> ### Author Response · Authors · 2025-09-29
> **Rebuttal by Authors (2)**
>
> > The impact of both the number of posteriors and the number of components per posterior is not deeply discussed, yet this information would be important for practitioners. Related to this, it would be insightful to analyze how stacking handles poor-quality posteriors and whether the weight optimisation effectively downweights them.
>
> The **number of posteriors** $M$ (obtained from independent VBMC runs) represents the main hyperparameter in S-VBMC. As a general rule of thumb, the more posteriors are stacked, the better the resulting S-VBMC approximation, as shown in Sections 4.4 and 4.5. However, as shown in Section 4.6, increasing the number of posteriors also increases computational costs. With this in mind, we made observations meant to serve as practical recommendations for potential practitioners in the paper. In Section 4.4 and 4.5, we point out that most of the gains in our metrics happen when we stacked up to 10 VBMC runs ($M = 10$), with relatively small additional improvements when stacking more. Furthermore, in Section 4.6, we discuss how stacking 10 runs vastly improves the posterior with relatively small ($\approx 5-15$ %) computational overhead. Therefore, our experiments showed $M=10$ to be a nice “sweet spot” where the posterior improves greatly and the overhead remains contained. We agree that this information should be re-iterated as an explicit recommendation in the Discussion (Section 6). We will add this.
>
> As for the **number of components per posterior**, this is set by the VBMC algorithm (Acerbi, 2018; 2020), and we used the default – and recommended – settings of the `pyvbmc` package (Huggins et al., 2023) for all our experiments (mentioned in Section 4.1). This results in a final number of 50 components per VBMC posterior, so that the total number of components for all our experiments was $M \times 50$. We will edit Section 4.1 to include this information.
>
> Regarding poor-quality posteriors, our initialisation strategy (Eq. 12) already downweights them by incorporating each run's ELBO into the initial weights. During optimisation, the algorithm further adjusts these weights based on each component's contribution to the stacked ELBO. In our experiments, we only filtered out VBMC runs which had been assessed as non-converged (as flagged by the `pyvbmc` software, Huggins et al., 2023) or poorly converged (i.e., with high uncertainty estimates of the components of the expected log-joint $I_k$ from Bayesian quadrature). We intentionally avoided filtering out low-ELBO VBMC posteriors (which might have converged to low-density posterior regions) to test S-VBMC's robustness. The consistent improvements we observed suggest that the weight optimisation effectively handles variability in posterior quality. In particular, “stray” posteriors with low ELBO would be automatically downweighted to have negligible weight in the mixture. We will add a brief discussion of this robustness property to the manuscript, and we will edit Section 3.1 (Procedure) to be more specific about our filtering procedure.
>
>
> > Finally, additional experiments in higher dimensions would be highly informative. As the authors acknowledge, dimensionality is a known limitation of VBMC. It would be interesting to see whether stacking can extend its applicability in this regard.
>
> We agree that exploring S-VBMC's behavior in higher dimensions would be valuable. However, the dimensional limitation stems from the underlying VBMC algorithm's reliance on Gaussian process surrogates, which face well-known scaling challenges beyond $\approx 10$ dimensions (Acerbi, 2018; 2020). While S-VBMC inherits this limitation, we hypothesise it might provide slight improvements in moderately high dimensions (e.g., $\approx 10$) by capturing different regions or aspects of the posterior through diverse initialisations. Still, this would likely not provide a silver bullet for issues in the underlying Gaussian process surrogate modelling of each individual VBMC run. Testing this hypothesis would require careful experimental design and is an interesting direction for future work. We will note this potential avenue in our discussion of future research directions.
>
>
> **References**
>
> Acerbi, L. (2018). Variational Bayesian Monte Carlo. Advances in Neural Information Processing Systems, 31, 8222-8232.
>
> Acerbi, L. (2019). An exploration of acquisition and mean functions in Variational Bayesian Monte Carlo. Proceedings of The 1st Symposium on Advances in Approximate Bayesian Inference (PMLR) 96, 1–10.
>
> Acerbi, L. (2020). Variational Bayesian Monte Carlo with noisy likelihoods. Advances in neural information processing systems, 33, 8211-8222.
>
> Huggins, B., Li, C., Tobaben, M., Aarnos, M. J., & Acerbi, L. (2023). PyVBMC: Efficient Bayesian inference in Python. Journal of Open Source Software 8(86), 5428.

---

> > ### Comment · Reviewer_ajye · 2025-10-03
> >
> > Most of the concerns raised in my review were adequately addressed in the authors’ response. An important clarification was that VBMC has a dynamic procedure to set the number of mixture components at runtime, with a maximum of 50 components, which directly addresses overlapping concerns raised by reviewers vbeX, yRB2, and myself. The exchange with reviewer 7953 also clarified Section 5, in particular that the identified bias mainly affects tasks such as model comparison rather than posterior approximation in general. Overall, the responses were thorough, and most of my questions were satisfactorily answered.

---

### Author Response · Authors · 2025-10-07
**Summary of changes by Authors**

We would like to thank each reviewer for their appreciative reviews and insightful comments. We edited and resubmitted our manuscript following their recommendations, and we believe it’s better for it. Here we list the changes we have made for the reviewers’ convenience.

- We moved Figure 1 (and the reference to it) to Section 1 (Introduction), following the advice of Reviewer 7953.
We added a section to Appendix A.3 (former A.2, see last point) with details about BBVI runtime in comparison to S-VBMC,  as suggested by Reviewer 7953.
- We expanded the first paragraph of Section 5 to explain the problem of the ELBO bias more clearly, with specific references to previous sections and figures, as suggested by Reviewer 7953.
- We added explicit recommendations for practitioners about the number of VBMC runs to stack in Section 6 (Discussion), as suggested by Reviewers 7953 and ajye.
- In Section 4.1 (Procedure), we clarified the fact that all our VBMC posteriors had 50 Gaussian components, as per default `pyvbmc` settings, prompted by comments by Reviewers 7953, yRB2, and ajye.
- We made the dimensionality limitations within which we operate ($D \leq 10$) clear from Section 1 (Introduction), as suggested by Reviewer  7953.
- We took out the denominator from the weight initialisation equation (now Equation 17), as it was incorrect, as Reviewer yRB2 pointed out.
- We added hardware specifications in Section 4.1 (Procedure), as requested by Reviewer yRB2.
- We corrected our notation, as it was confusing, as Reviewer vBex pointed out. Now we use the hat notation (e.g., $\hat{I}_k$) for *VBMC’s estimates*, and the hatless ones (e.g., $I_k$) for the (unknown) true values.
- We explained how we re-parametrised the weights of the stacked posterior with unconstrained logits, as this information was previously absent, as noted by Reviewers vBex and ajye. In doing this, we slightly rearranged the order of Section 3 (S-VBMC) to improve its flow.
- We added a clarifying sentence under former Equation 28 (now Equation 30), as suggested by Reviewer vBex.
- Prompted by comments from Reviewer ajye, we added a short discussion about the robustness of S-VBMC when it comes to low-ELBO VBMC posteriors, as well as one about potential experiments with problems with higher dimensionality. Both are found in Section 6 (Discussion).
- Prompted by a comment from Reviewer ajye, we added details about our filtering procedures in Section 3.1 and added a section to Appendix A4 (former A.3, see last point) to present the results of those filters.
- After internal discussion, we decided that the paper was missing some information about our implementation, which, while not essential to understand the approach and experiments, is important to ensure reproducibility. This has been added as Appendix A.2 and is referred to in the relevant sections of the main text (Sections 3.1 and 5.1).

---

### Decision · Action_Editor_LnUv · 2025-10-19

**Recommendation:** Accept as is

**Audience:**

Yes

**Audience Explanation:**

All reviewers agree that a large portion of TMLR's audience would find this work interesting.

**Claims And Evidence:**

Yes

**Claims Explanation:**

All four reviewers affirmed that the manuscript's claims are supported by evidence, ultimately recommending acceptance. The primary accuracy concerns identified were technical rather than fundamental: the authors corrected an incorrect denominator in the weight initialization equation alongside some confusing notation and missing details about the reparametrization of weights with unconstrained logits. After the authors addressed these technical corrections and clarifications in their revision, all reviewers confirmed the manuscript was sound and accurate.